# AVERAGE SENSITIVITY OF DECISION TREE LEARNING

**Satoshi Hara**
Osaka University
satohara@ar.sanken.osaka-u.ac.jp

**Yuichi Yoshida**
National Institute of Informatics
yyoshida@nii.ac.jp

## ABSTRACT

A decision tree is a fundamental model used in data mining and machine learning. In practice, the training data used to construct a decision tree may change over time or contain noise, and a drastic change in the learned tree structure owing to such data perturbation is unfavorable. For example, in data mining, a change in the tree implies a change in the extracted knowledge, which raises the question of whether the extracted knowledge is truly reliable or is only a noisy artifact. To alleviate this issue, we design decision tree learning algorithms that are stable against insignificant perturbations in the training data. Specifically, we adopt the notion of average sensitivity as a stability measure, and design an algorithm with low average sensitivity that outputs a decision tree whose accuracy is close to the optimal decision tree. The experimental results on real-world datasets demonstrate that the proposed algorithm enables users to select suitable decision trees considering the trade-off between average sensitivity and accuracy.

## 1 INTRODUCTION

A decision tree is a fundamental model in applications such as extracting knowledge in data mining and predicting outcomes in machine learning. Learned decision trees enable the extraction of hidden structures in the data in an interpretable manner using the if-then format. In data mining, the extracted structures are of fundamental interest (Rokach & Maimon, 2007; Gorunescu, 2011). Decision trees also play an essential role in decision making (Zeng et al., 2017; Rudin, 2019; Arrieta et al., 2020) because unlike complex models, such as deep neural networks, the decisions made by decision trees are explainable. With the increase of the utility of machine learning models in real-world problems, decision trees and their variants are widely used particularly for applications such as high-stake decision making, where explainability is crucial and transparency higher than post-hoc explanations (e.g., (Angelino et al., 2018; Rudin, 2019; Arrieta et al., 2020)) are required.

Current studies on decision trees and their families mainly focus on developing learning algorithms to improve two aspects of learned trees: accuracy and interpretability. Here, we demonstrate that there is a third essential aspect that is missing in current studies: the *stability* of the learning algorithm against insignificant perturbations on the training data. Decision trees are typically used to extract knowledge from data and help users make decisions that can be explained. If the learning algorithm is unstable, the structure of the learned trees can vary significantly even for insignificant changes in the training data. In data mining, this implies that the extracted knowledge can be unstable, which raises the question of whether the extracted knowledge is truly reliable or only a noisy artifact induced by the unstable learning algorithm. In model-based decision making, this implies that the decision process can change drastically whenever a few additional data are obtained and the tree is retrained on the new training data. Such noisy decision makers are unacceptable for several reasons. For example, stakeholders may lose their trust in such decision makers, or it may be extremely costly to frequently and drastically update the entire decision making system.

Figure 1 shows an illustrative example of sensitive/stable decision tree learning algorithms. In this example, the standard greedy tree learning algorithm induces different trees before and after one data point (large red triangle) is removed (Figure 1(a)). Thus, it can be observed that the greedy algorithm is sensitive to the removal of data points. The objective of this study is to design a tree learning algorithm that can induce (almost) same trees against the removal of a few data points (Figure 1(b)).

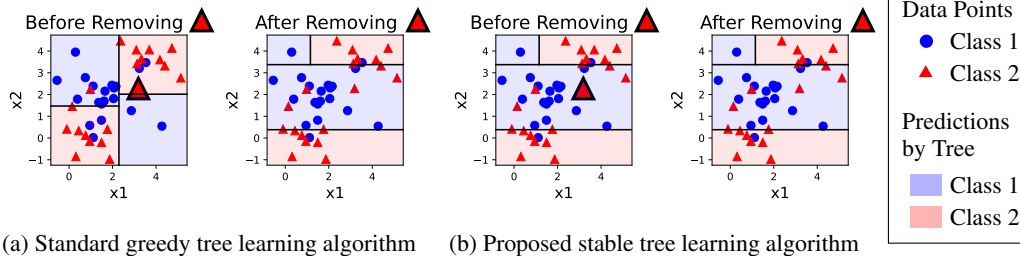

Figure 1: Decision boundaries of the learned decision trees. (a) The standard greedy tree learning algorithm is sensitive to the removal of even a single data point (large red triangle) from the training data. (b) The proposed learning algorithm produces more stable trees.

In this study, we design a decision tree learning algorithm that is stable against insignificant perturbations in the training data. Specifically, we consider the change in (the distribution of) the learned tree upon deletion of a random data point from the training data, using the notion of *average sensitivity* (Varma & Yoshida, 2021). Subsequently, we design a (randomized) decision tree learning algorithm with low average sensitivity while preserving the accuracy of the learned decision tree up to a tolerance parameter.

A randomized algorithm may output completely different decision trees on the original training data and on the training data obtained by deleting a random data point even if the output distributions are close. To alleviate this issue, we design a (randomized) decision tree learning algorithm with low expected average sensitivity over random bits used in the algorithm, which implies that the output decision tree on the original training data and that on the training data obtained by deleting a random data point are close with a high probability over the choice of the random bits used.

Through real-world data experiments, we demonstrate that our learning algorithm exhibits a lower average sensitivity compared to the standard greedy decision tree learning algorithm, while maintaining the decrease in accuracy within the prescribed tolerance parameter.

## 2 RELATED WORK

**Decision Tree Learning Algorithms** Generally, learning an optimal decision tree is NP-hard (Laurent & Rivest, 1976), and hence we can obtain optimal trees only for small problems (Bertsimas & Dunn, 2017; Angelino et al., 2018; Günlük et al., 2021). To avoid this issue, recursive greedy splitting is widely used for learning (non-optimal) decision trees (Rivest, 1987; Loh, 2011; Quinlan, 2014), and Bayesian approaches are used to learn a family of decision trees, such as rule lists and rule sets (Wang et al., 2017; Yang et al., 2017). These studies are concerned with learning trees with less computation or better interpretability. This study is orthogonal to them in that our interest is developing stable decision tree learning algorithms, which was not considered before. We stress that the focus of the current study is to learn a stand-alone decision tree and not to learn a collection of decision trees for ensemble models (e.g., (Ho, 1995; Breiman, 2001; Friedman, 2001; Chen & Guestrin, 2016)). For the latter, we want decision trees that make different predictions, and hence somewhat sensitive algorithms are favorable rather than stable ones.

**Average Sensitivity** Varma & Yoshida (2021) introduced the notion of average sensitivity and designed algorithms with low average sensitivity for various graph problems including the minimum spanning tree, minimum cut, and minimum vertex cover problems. Average sensitivity of algorithms are discussed also for various problems including the maximum matching problem (Yoshida & Zhou, 2021), problems that can be solved by dynamic programming (Kumabe & Yoshida, 2022a;b), spectral clustering (Peng & Yoshida, 2020), and Euclidean $k$-clustering (Yoshida & Ito, 2022).

**Adversarial Robustness** Insignificant human-imperceptible perturbations to the input can mislead trained models, and such perturbations are called *adversarial attacks*. It is known that adversarial attacks are harmful to decision trees (Chen et al., 2019a;b; Kantchelian et al., 2016). To alleviate this issue, several recent studies have considered the problems of robustness verification (Chen et al., 2019b; Törnblom & Nadjm-Tehrani, 2019; Wang et al., 2020) and adversarial defense (Chen et al., 2019a; Andriushchenko & Hein, 2019; Calzavara et al., 2020; Chen et al., 2021). Adversarial at-

tacks focus on the change in the predicted label against perturbing a single input data point at the inference time, whereas average sensitivity focuses on the change in the learned decision tree against perturbing the entire training data at the learning time.

**Stability** Bousquet and Elisseeff introduced the notion of the stability of a learning algorithm and discussed its relation to generalization ability (Bousquet & Elisseeff, 2002). Unlike average sensitivity, this notion only concerns the stability of the loss value against data perturbation. Hence, it cannot be used to stabilize the structure of learned decision tree.

**Differential Privacy** Differential privacy (Dwork, 2006) measures the stability of the output against perturbation to the input. Decision trees have been intensively studied from the perspective of differential privacy (see (Fletcher & Islam, 2019) and references therein). The average sensitivity of an algorithm can be bounded by the differential privacy parameter $\epsilon$ times the maximum size of the output (Varma & Yoshida, 2021). In the decision tree learning setting, the bound we obtain in this manner is roughly $O(\epsilon \cdot 2^B)$, where $B$ specifies the depth of the output decision tree. By contrast, our bound (Theorem 4.1) is $O(B2^B/n)$, where $n$ is the number of points in the training data. Our bound is much smaller because we usually set $2^B \ll n$ to avoid overfitting.

## 3 PRELIMINARIES

### 3.1 DECISION TREE

Let $\mathcal{X}$ and $\mathcal{Y}$ be input and output spaces, respectively. We call a Boolean function $\omega : \mathcal{X} \to \{0, 1\}$ a *decision rule*. Then, a *decision tree* is a function $\phi : \mathcal{X} \to \mathcal{Y}$ represented by a rooted proper binary tree, that is, it has a special node called the *root* and each node has either zero or two children, such that each internal node $t$ is associated with a decision rule $\omega_t : \mathcal{X} \to \{0, 1\}$ and each leaf is associated with a label $y_t \in \mathcal{Y}$. Then given an input $x \in \mathcal{X}$, the decision tree $\phi$ predicts a label $y$ according to PREDICT($\phi, x$) shown in Algorithm 1.

Let $\mathcal{L} = ((x_1, y_1), \dots, (x_n, y_n))$ be a training data. The *total score* of a decision tree $\phi : \mathcal{X} \to \mathcal{Y}$ with respect to $\mathcal{L}$ is defined to be $s(\phi, \mathcal{L}) := \sum_{i=1}^{n} 1[\phi(x_i) = y_i]$, where $1[X]$ is the indicator of the event $X$. Note that $s(\phi, \mathcal{L})/n$ is the accuracy of $\phi$ on $\mathcal{L}$.

For a nonnegative integer $B$, let $\mathcal{T}_B$ be the set of decision trees of depth $B$. Then, let $\mathrm{opt}_B(\mathcal{L}) := \max_{\phi \in \mathcal{T}_B} s(\phi, \mathcal{L})$ be the maximum total score of a decision tree of depth $B$. Also for a decision rule $\omega : \mathcal{X} \to \{0, 1\}$ and a nonnegative integer $B$, let $\mathcal{T}_{\omega, B}$ be the set of decision trees of depth $B$ with the root node having the decision rule $\omega$. Then we define $\mathrm{opt}_{\omega, B}(\mathcal{L}) := \max_{\phi \in \mathcal{T}_{\omega, B}} s(\phi, \mathcal{L})$. We clearly have $\mathrm{opt}_B(\mathcal{L}) = \max_\omega \mathrm{opt}_{\omega, B}(\mathcal{L})$.

For a decision tree $\phi$, we denote by $|\phi|$ the number of nodes (including the leaves) in $\phi$. For two decision trees $\phi$ and $\phi'$, we define the distance $d_{\mathrm{DT}}(\phi, \phi')$ between them as the output of DISTANCE($\phi, \phi'$) shown in Algorithm 2. Intuitively, the procedure DISTANCE($\phi, \phi'$) computes the maximal subtree common to $\phi$ and $\phi'$, and then outputs the total number of remaining nodes after subtracting the common subtree from $\phi$ and $\phi'$. It is clear that $d_{\mathrm{DT}}(\cdot, \cdot)$ satisfies triangle inequality.

We note that, even if two decision trees $\phi, \phi' : \mathcal{X} \to \mathcal{Y}$ are equal as a function, they may have a large distance in $d_{\mathrm{DT}}$ if they have different tree structures. For interpretability, however, we believe this is the advantage of using $d_{\mathrm{DT}}$ because it is not easy to verify the equivalence of $\phi$ and $\phi'$ when they have different tree structures and $d_{\mathrm{DT}}$ correctly reflect this situation.

### 3.2 AVERAGE SENSITIVITY

For a training data $\mathcal{L} = ((x_1, y_1), \dots, (x_n, y_n))$ and $1 \leq i \leq n$, let $\mathcal{L}^{(i)} = ((x_1, y_1), \dots, (x_{i-1}, y_{i-1}), (x_{i+1}, y_{i+1}), \dots, (x_n, y_n))$ be the training data obtained from $\mathcal{L}$ by dropping the $i$th data point.

Let $A$ be a deterministic algorithm that, given a training data, outputs a decision tree. Then, the *average sensitivity* of $A$ on $\mathcal{L}$ is

$$\frac{1}{n} \sum_{i=1}^{n} d_{\mathrm{DT}}(A(\mathcal{L}), A(\mathcal{L}^{(i)})).$$

---

**Algorithm 1:**

---

1 **Procedure** PREDICT($\phi, x$)
2     Let $t$ be the root node of $\phi$;
3     **if** $t$ *is a leaf* **then return** $y_t$;
4     **else**
5         Let $\phi_L, \phi_R$ be the decision trees rooted at the left and right children of $t$, respectively;
6         **if** $\omega_t(x) = 0$ **then return** PREDICT($\phi_L, x$);
7         **else return** PREDICT($\phi_R, x$);

---

**Algorithm 2:**

---

1 **Procedure** DISTANCE($\phi, \phi'$)
2     Let $t$ and $t'$ be the root nodes of $\phi$ and $\phi'$, respectively;
3     **if** *both $t$ and $t'$ are leaves* **then**
4         **return** $0$ if $y_t = y_{t'}$ and $2$ otherwise.
5     **else if** *either $t$ or $t'$ is a leaf* **then return** $|\phi| + |\phi'|$;
6     **else if** $\omega_t \neq \omega_{t'}$ **then return** $|\phi| + |\phi'|$;
7     **else**
8         Let $\phi_L, \phi_R$ be the decision trees rooted at the left and right children of $t$, respectively;
9         Let $\phi'_L, \phi'_R$ be the decision trees rooted at the left and right children of $t'$, respectively;
10         **return** DISTANCE($\phi_L, \phi'_L$) + DISTANCE($\phi_R, \phi'_R$).

---

Now, we generalize this definition for randomized algorithms. First for two distributions $\mathcal{D}$ and $\mathcal{D}'$ over decision trees, we define their *earth mover's distance* as $d_{\mathrm{EM}}(\mathcal{D}, \mathcal{D}') = \min_{\mathcal{P}} \mathbf{E}_{(\phi,\phi') \sim \mathcal{P}} d_{\mathrm{DT}}(\phi, \phi')$, where $\mathcal{P}$ is over distributions of pairs of decision trees such that the marginal distributions on the first and second coordinates are equal to $\mathcal{D}$ and $\mathcal{D}'$, respectively. Let $A$ be a randomized algorithm that, given a training data, outputs a decision tree. Then, we define the *average sensitivity* of $A$ on $\mathcal{L}$ as

$$\frac{1}{n} \sum_{i=1}^{n} d_{\mathrm{EM}}(A(\mathcal{L}), A(\mathcal{L}^{(i)})), \tag{1}$$

where we regard $A(\cdot)$ as a distribution over decision trees.

It is natural to consider a variant of average sensitivity such that we delete $k$ data points instead of a single data point because in practice many data points can be dropped. As discussed in Varma & Yoshida (2021), this variant can be bounded by $k$ times the average sensitivity above, and hence we focus on bounding the latter.

## 4   DECISION TREE CONSTRUCTION

In this section, we provide a decision tree learning algorithm with a high total score and a low average sensitivity. Specifically, we show the following:

**Theorem 4.1.** *There exists an (possibly inefficient) randomized algorithm that, given a training data $\mathcal{L}$ of size $n$, a depth bound $B$, and a parameter $\epsilon > 0$, returns a decision tree $\phi$ of depth at most $B$ such that $\mathbf{E}_{\phi}[s(\phi, \mathcal{L})] \geq (1 - \epsilon)^B \mathrm{opt}_B(\mathcal{L})$, and for the set of decision rules $\Omega$, its average sensitivity is $O\left(\frac{B 2^B \log |\Omega|}{\epsilon n}\right)$.*

We note that we usually choose $2^B \ll n$ to avoid overfitting and $\epsilon = \Theta(1/B)$ to achieve constant approximation, and hence the average sensitivity bound is essentially $O(1/n)$. We note that the algorithm of Theorem 4.1 makes use of the optimal total score, which is not efficiently computable in general. We discuss practical implementations of the algorithm in Section 6.

The algorithm of Theorem 4.1 is based on a procedure called STABLEDR, which selects a decision rule in a stable way (Algorithm 3). This is a simple application of the exponential mechanism (Mc-

---

**Algorithm 3:**

1  **Procedure** STABLEDR($\mathcal{L}, B, \epsilon$)

2      Select $\omega \in \Omega$ with probability $\propto \exp(\lambda \cdot \mathrm{opt}_{\omega,B}(\mathcal{L}))$, where $\lambda = \frac{2 \log |\Omega|}{\epsilon \cdot \mathrm{opt}_B(\mathcal{L})}$ ;

3      **return** $\omega$.

4  **Procedure** STABLEDT'($\mathcal{L}, B, \epsilon, d$)

5      **if** $|\mathcal{L}| \leq 1$ *or* $d = B$ **then return** an optimal label for $\mathcal{L}$;

6      $\omega \leftarrow$ STABLEDR($\mathcal{L}, B, \epsilon$);

7      Partition $\mathcal{L}$ into $\mathcal{L}_L \cup \mathcal{L}_R$ according to $\omega$;

8      $\phi_L \leftarrow$ STABLEDT'($\mathcal{L}_L, B, \epsilon, d+1$) and $\phi_R \leftarrow$ STABLEDT'($\mathcal{L}_R, B, \epsilon, d+1$);

9      Let $\phi_\omega$ be the decision tree such that the root node $t$ has rule $\omega$ and the left and right children of $t$ are $\phi_L$ and $\phi_R$, respectively;

10     **return** $\phi_\omega$.

11  **Procedure** STABLEDT($\mathcal{L}, B, \epsilon$)

12     **return** STABLEDT'($\mathcal{L}, B, \epsilon, 0$).

---

Sherry & Talwar, 2007) to $\mathrm{opt}_{\omega,B}(\mathcal{L})$ ($\omega \in \Omega$). Then, the proposed algorithm, STABLEDT (Algorithm 3), works as follows. At each node, we compute a decision rule $\omega$ using STABLEDR, split the training data according to $\omega$, and recursively construct decision trees for each of the split data until the size of the training data becomes at most one or the depth becomes $B$.

To understand why the randomized procedure STABLEDR produces stability, consider the following scenario: Suppose that we have two candidate rules $\omega_1$ and $\omega_2$ with optimal scores $\mathrm{opt}_{\omega_1,B}(\mathcal{L}) = 90$ and $\mathrm{opt}_{\omega_2,B}(\mathcal{L}) = 89$, respectively. The greedy algorithm selects $\omega_1$ because it has the larger score. Suppose that the scores have changed upon the removal of a subset $\mathcal{S} \subset \mathcal{L}$ and now we have $\mathrm{opt}_{\omega_2,B}(\mathcal{L} \setminus \mathcal{S}) = 85$ and $\mathrm{opt}_{\omega_2,B}(\mathcal{L} \setminus \mathcal{S}) = 86$. Then, the greedy algorithm selects $\omega_2$ because it has now the larger score, and hence we never obtain the same tree for $\mathcal{L}$ and $\mathcal{L} \setminus \mathcal{S}$. By contrast, the probability that STABLEDR selects $\omega_1$ (resp., $\omega_2$) is nearly half for both $\mathcal{L}$ and $\mathcal{L} \setminus \mathcal{S}$ because $\omega_1$ (resp., $\omega_2$) is a nearly optimal rule. Hence, the distributions of the trees given by Algorithm 3 are close between $\mathcal{L}$ an $\mathcal{L} \setminus \mathcal{S}$.

## 5   EXPECTED DETERMINISTIC AVERAGE SENSITIVITY

Recall that the average sensitivity (1) for randomized algorithms bounds the earth mover's distance between the distributions of $A(\mathcal{L})$ and $A(\mathcal{L}^{(i)})$. This does not immediately imply that, given a decision tree for $\mathcal{L}$, we can compute a similar decision tree for $\mathcal{L}^{(i)}$ with a similar total score because the mapping from decision trees for $\mathcal{L}$ to those for $\mathcal{L}^{(i)}$ that achieves the earth mover's distance is not always available. To address this issue, we consider bounding the expectation of deterministic average sensitivity between output decision trees over random bits:

$$\mathbf{E}_\pi \left[ \frac{1}{n} \sum_{i=1}^{n} d_{\mathrm{DT}}(A_\pi(\mathcal{L}), A_\pi(\mathcal{L}^{(i)})) \right], \tag{2}$$

where $A_\pi$ is the *deterministic* algorithm obtained from a randomized algorithm $A$ by fixing the random bits used in $A$ to $\pi \in \{0, 1\}^*$. Then, using the same random bits $\pi$, we can obtain similar decision trees for $\mathcal{L}$ and $\mathcal{L}^{(i)}$. Note that the average sensitivity (1) is bounded from above by the expected deterministic average sensitivity (2) because the pair of random variables $(A_\pi(\mathcal{L}), A_\pi(\mathcal{L}^{(i)}))$ induces a joint distribution over pairs of decision trees.

By slightly modifying STABLEDT and the proof of Theorem 4.1, we obtain the following:

**Theorem 5.1.** *There exists a (possibly inefficient) randomized algorithm that, given a training data $\mathcal{L}$ of size $n$, a depth bound $B$, and a parameter $\epsilon > 0$, returns a decision tree $\phi$ of depth at most $B$ such that $\mathbf{E}_\phi[s(\phi, \mathcal{L})] \geq (1 - \epsilon)^B \mathrm{opt}_B(\mathcal{L})$, and for the set of decision rules $\Omega$, its expected average sensitivity over random bits is $O\left( \frac{B 2^B \log |\Omega|}{\epsilon n} \right)$.*

Table 1: Datasets

| | Dataset | training data size | sampled data size | test data size | # of features | # of classes | tree depth |
|---|---|---|---|---|---|---|---|
| small | breast cancer | 546 | 436 | 137 | 10 | 2 | 5 |
| | diabetes | 614 | 491 | 154 | 8 | 2 | 1 |
| large | cod-rna | 59535 | 1000 | 271617 | 8 | 2 | 10 |
| | covtype | 400000 | 1000 | 181000 | 54 | 7 | 7 |
| | higgs | 10500000 | 1000 | 500000 | 28 | 2 | 3 |
| | ijcnn | 49900 | 1000 | 91701 | 22 | 2 | 1 |
| | sensorless | 48509 | 1000 | 10000 | 48 | 11 | 9 |
| | webspam | 300000 | 1000 | 50000 | 254 | 2 | 7 |

## 6 PRACTICAL IMPLEMENTATIONS

The algorithm STABLEDR and the algorithm of Theorem 5.1 require the value of $\mathrm{opt}_{\omega,B}(\mathcal{L})$, which is NP-hard to compute (Laurent & Rivest, 1976). Although several algorithms based on integer programming were proposed (Bertsimas & Dunn, 2017; Günlük et al., 2021), they are not fast enough to handle large data. To address this issue, we suggest replacing $\mathrm{opt}_{\omega,B}(\mathcal{L})$ with $\mathrm{opt}_{\omega,1}(\mathcal{L})$ because the latter can be computed quite efficiently. Note that this strategy is the same as the standard greedy recursive splitting used for training (non-optimal) decision trees (Rivest, 1987; Loh, 2011; Quinlan, 2014). We use this version in our experiments in Section 7. The same analysis goes through and we obtain the following.

**Theorem 6.1.** *There exists a randomized algorithm that, given a training data $\mathcal{L}$ of size $n$, a depth bound $B$, and a parameter $\epsilon > 0$, returns a decision tree $\phi$ of depth at most $B$ such that $\mathbf{E}[s(\phi, \mathcal{L})] \geq (1 - \epsilon)\mathrm{opt}_1(\mathcal{L})$, and for the set of decision rules $\Omega$, its expected average sensitivity over random bits is $O\left(\frac{B2^B \log|\Omega|}{\epsilon n}\right)$. The time complexity is $\sum_{d=0}^{\lceil \log n \rceil - 1} 2^B \cdot T(n/2^B) \cdot |\Omega|$, where $T(m)$ is the running time required to compute $\mathrm{opt}_{\omega,1}(\mathcal{L}')$ for $\omega \in \Omega$ with $|\mathcal{L}'| = m$.*

We note that, although the approximation guarantee of Theorem 6.1 is weaker than those of Theorems 4.1 and 5.1, practically used decision tree learning algorithms also lack approximation guarantees. In Section 7, we empirically confirm that the accuracy of the algorithm of Theorem 6.1 is not much worse than those of baseline algorithms. We also note that, in many cases, computing $\mathrm{opt}_{\omega,1}(\mathcal{L}')$ takes linear time, and sometimes we can compute $\mathrm{opt}_{\omega,1}(\mathcal{L}')$ for many $\omega$'s at once.

## 7 EXPERIMENTS

We demonstrate that the proposed algorithm can output stable decision trees.

### 7.1 SETUPS

**Datasets** We used datasets shown in Table 1.[1] We split the datasets into small ones (breast cancer and diabetes) and large ones (cod-rna, covtype, higgs, ijcnn, sensorless, and webspam). We use small datasets to demonstrate the stability of the proposed algorithm, and large datasets to study trade-offs between average sensitivity and accuracy. In the experiments, we used subsamples of these datasets. For training, we randomly sampled 80% of the data points and 1000 data points for small and large datasets, respectively. In the experiments, we evaluated the test accuracy of the learned decision trees using the entire test data.

**Tree Learning Algorithms** In the experiment, we used the practical implementation described in Section 6 for training the decision trees. As we have shown in Theorem 6.1, this practical implementation can output stable decision trees without solving NP-hard problems. We also adopted the following greedy tree learning algorithm as the baseline for comparison. At each node, the greedy

---

[1]These datasets are obtained from `https://github.com/chenhongge/RobustTrees` We omitted MNIST-based datasets because decision trees are typically not used for images.

algorithm selects the decision rule $\omega$ with the highest score, i.e., $\omega \in \arg\max_{\omega \in \Omega} \mathrm{opt}_{\omega,1}(\mathcal{L})$. When multiple decision rules exist in $\arg\max_{\omega \in \Omega} \mathrm{opt}_{\omega,1}(\mathcal{L})$, we select one of them uniformly at random. We set the tree depth shown in Table 1 so that the greedy algorithm exhibits the highest accuracy in cross-validation.

We implemented both the greedy and proposed algorithms in Python 3 using the JIT compiler of Numba. We used the equidistant points $\{x_{j,1}, \ldots, x_{j,Q}\}$ within the interval $[x_{j,\min}, x_{j,\max}]$ for each feature $x_j$ as the decision rules, where $x_{j,\min} = x_{j,1}$ and $x_{j,\max} = x_{j,Q}$ are the minimum and the maximum of $x_j$ in the dataset and we set $Q = 500$. That is, we set $\Omega = \{u \mapsto \mathbb{1}[u_j \le x_{j,q}]\}_{j,q}$,

**Procedure** We generated 10 sampled training data from the original training data. For each of the sampled training data, we trained decision trees using the greedy algorithm and proposed algorithms over different values of $\epsilon$. Because the tree learning algorithms are randomized, we trained trees for ten different random bits $\pi_1, \pi_2, \ldots, \pi_{10}$. Through this procedure, we obtained 100 trees for each tree learning algorithm (over ten sampled training data and ten different random bits). We report the average of the average sensitivity and the training and test accuracy over these 100 trees. To estimate the average sensitivity, we trained additional trees using the sampled training data, with $m$ data points removed at random. Formally, let $A(\mathcal{L})$ be the tree trained using the algorithm $A$ on $\mathcal{L}$. Then, we consider another tree $A(\mathcal{L} \setminus \mathcal{S})$ trained on the set $\mathcal{L} \setminus \mathcal{S}$ with $|\mathcal{S}| = m$. We selected the sets $\mathcal{S}_1, \mathcal{S}_2, \ldots, \mathcal{S}_R$ at random, and estimated the (normalized) expected deterministic average sensitivity over random bits as $\frac{1}{R}\sum_{r=1}^{R} \frac{d_{\mathrm{DT}}(A_{\pi_t}(\mathcal{L}), A_{\pi_t}(\mathcal{L}\setminus\mathcal{S}_r))}{d_B}$, where $\pi_t$ is the $t$-th random bits, and $d_B = 2^{B+2} - 2$ is the maximum distance between the trees with depth $B$, which normalizes the sensitivity within $[0, 1]$. We interpret this normalized sensitivity as the fraction of different nodes between the two trees. We set $R = 100$, and varied the number $m$ of removed data points from 1 to 30% of the sampled training data.

## 7.2 RESULT 1: DEPENDENCY ON $\epsilon$

First, we evaluated the dependency of the proposed algorithm on the choice of $\epsilon$ for small datasets. In the experiment, we evaluated the algorithm over nine different $\epsilon$ values from $10^{-2}$ to $10^0$ on a logarithmic scale and over the number of removed data points $m = 1$, 1%, and 10% of the sampled training data. Figure 2 shows the average sensitivity, training accuracy, and test accuracy. It is evident from Figures 2 (a) and (d) that the average sensitivity decreases as $\epsilon$ increases as suggested by Theorem 6.1. Empirically, we observed that the average sensitivity is kept almost constant until $\epsilon = 0.1$ and it began to decrease for larger $\epsilon$ values. Figures 2 (b) and (e) show that the training accuracy decreases for $\epsilon$ larger than 0.1. These results suggest that the stability of the trees is obtained at the cost of the decrease in training accuracy. This observation is consistent with Theorem 6.1 as well. Based on these results, we can empirically confirm the correctness of Theorem 6.1: the trained decision trees become stable, particularly for larger $\epsilon$, whereas the accuracy decrease. We also note that Figures 2(c) and (f) suggest that this may not always be the case for test accuracy, which is outside the scope of our theorem.

## 7.3 RESULT 2: EXAMPLES OF TREES

Figure 3 shows examples of the trees trained on the breast cancer dataset using the greedy and the proposed algorithms with $\epsilon = 0.3$ and $m = 10\%$ of the sampled training data. In the experiment, we trained 100 trees on the set $\mathcal{L} \setminus \mathcal{S}_r$ for $r = 1, \ldots, 100$ to estimate the expected average sensitivity over random bits. Within these 100 trees, some had identical structures, while others did not. In the figure, we show the original tree trained using all the sampled training data, and the three most frequently identical structures within the 100 trees. In this example, the original tree and the most frequent trees were identical.

There are two important implications in Figure 3. Firstly, the second frequent trees of the greedy algorithm changed drastically from the original tree, whereas in the proposed algorithm, the changes appeared only in small substructures. These smaller changes in the proposed algorithm induced a smaller distance $d_{\mathrm{DT}}(A(\mathcal{L}), A(\mathcal{L} \setminus \mathcal{S}_r))$, resulting in smaller expected deterministic average sensitivity for $\epsilon > 0.1$, as shown in Figure 2.

Secondly, the most frequent trees dominated 10/100 and 72/100 of the cases for the greedy and proposed algorithms, respectively. Therefore, the tree can change frequently upon data point removal

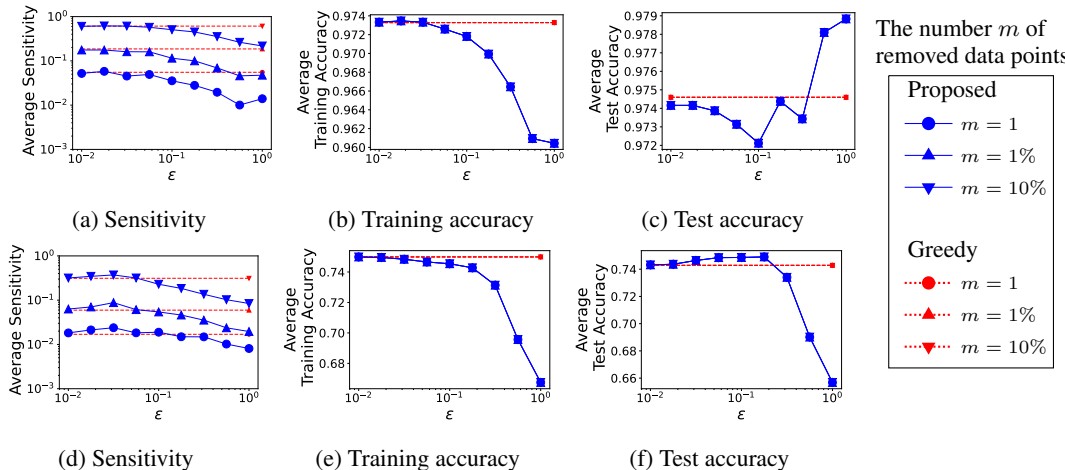

Figure 2: Average sensitivity and accuracy of the trained trees over different $\epsilon$ on small datasets, breast cancer (a)–(c) and diabetes (d)–(e). The blue and red lines denote the results for the proposed and greedy tree learning algorithms, respectively.

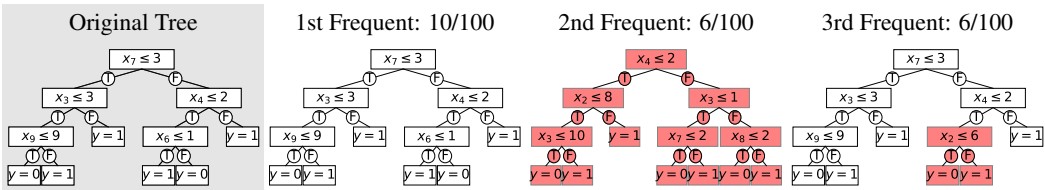

(a) (Greedy) The original tree (leftmost) and the first three frequent trees (the others).

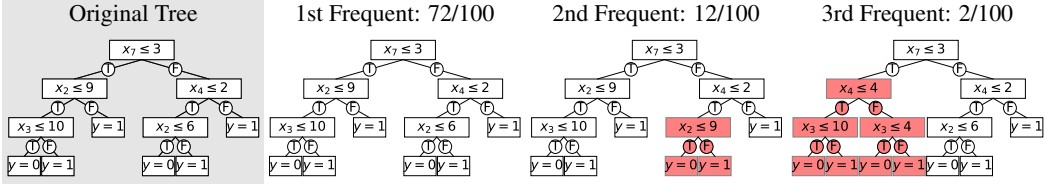

(b) (Proposed) The original tree (leftmost) and the first three frequent trees (the others).

Figure 3: The three most frequent trees on the breast cancer dataset for (a) the greedy and (b) proposed tree learning algorithms over 100 random data point removals with $\epsilon = 0.3$ and $m = 10\%$. "T" and "F" on the edges denote the splitting whether the rule in the parent node is True and False, respectively. In both methods, the most frequent trees are identical to the original tree trained using all the sampled training data. The red nodes denote differences from the original tree.

in the greedy algorithm, whereas the change is less frequent in the proposed algorithm, where more than 70% of the trees are identical. These results indicate that the proposed algorithm can induce trees with small distances and less frequent structural changes. This property is favorable in practical situations because in data mining applications, the extracted knowledge is guaranteed to not change drastically, and in machine learning applications, we can continue using almost the same decision-making process.

## 7.4 RESULT 3: SENSITIVITY-ACCURACY TRADE-OFF

The results in Figure 2 suggest that the average sensitivity and accuracy are in a trade-off relation through $\epsilon$. Using a large $\epsilon$, we obtain stable trees with a slight decrease in accuracy, where using a small $\epsilon$, we obtain accurate trees; however they tend to have high sensitivities.

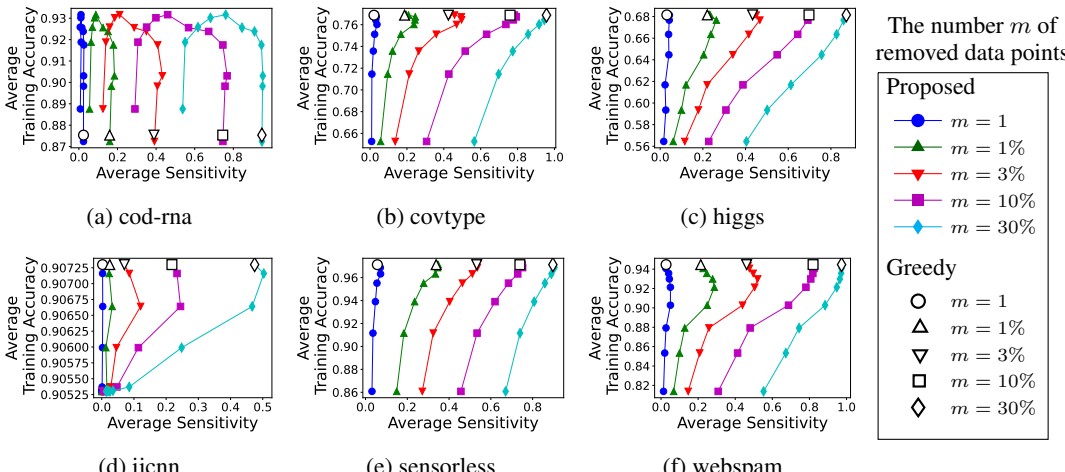

Figure 4: Trade-off curves between average sensitivity and training accuracy when $\epsilon$ is changed. We varied the number of training data points to be removed from one to 30% of the sampled training data. White markers denote the results of the greedy tree learning.

We computed this trade-off for large datasets by varying $\epsilon$ from $10^{-5}$ to $10^0$ on a logarithmic scale. Figure 4 shows the trade-off curves between the average sensitivity and training accuracy. In the figures, we observe that the average sensitivity and accuracy are in a trade-off relation for most of the datasets. The trees become stable as $\epsilon$ increases, while incurring an accuracy decrease in the bottom left of the figures. We note that our theory only guarantees that the decrease in training accuracy is limited. However, the test accuracy results in Figure 7 in Appendix D also confirm that the drop in test accuracy is limited. Thus, it is not necessary to sacrifice a considerable amount of the test accuracy to obtain stable decision trees in practice.

The result on cod-rna is an exception that did not exhibit clear trade-off curves. In this dataset, the trees become less stable and less accurate simultaneously for small $\epsilon$, as shown in the bottom right of the figures. In particular, the greedy trees tend to be less accurate than stable ones. One possible reason of this phenomenon is overfitting. In Table 1, the tree depth is set to ten for this dataset. In such deep trees, the number of training data points that reach a deep node tends to be small. Then, the greedy selection of the best decision rule based on a small number of data points can be unstable, some arbitrary rules may be selected by chance, and overfitting occurs. Therefore, the greedily selected rule is not optimal if we consider deeper trees, i.e., $\mathrm{opt}_{\omega,B}(\mathcal{L})$, can be smaller than some $\mathrm{opt}_{\omega',B}(\mathcal{L})$ even if $\mathrm{opt}_{\omega,1}(\mathcal{L}) > \mathrm{opt}_{\omega',1}(\mathcal{L})$. On the contrary, the proposed algorithm randomizes the choice of the decision rule using an exponential mechanism. We conjecture that this randomization can help prevent suboptimal rules and overfitting, and increase the chance of gaining more accurate trees.

Finally, we emphasize that the results confirm that we gained the freedom to choose decision trees on these trade-off curves using the proposed algorithm. This will open up a new paradigm for practitioners who have suffered from the high sensitivity of existing learning algorithms. Practitioners can now tune $\epsilon$ and obtain stable trees upon their demand.

# 8 CONCLUSION

We proposed decision tree learning algorithms with theoretical guarantees on its average sensitivity and accuracy. We experimentally confirmed that the proposed algorithms performed well on real-world datasets. An obvious open problem is to show some hardness on the trade-off between average sensitivity and accuracy. In practice, it is important that the predictions made by the learned model do not change significantly with slight changes in the training data. Hence, it is also natural to measure the distance between models by the number of mismatches of their predictions and study average sensitivity in terms of this distance even for "non-explainable" learning models such as random decision forests Ho (1995); Breiman (2001) and deep neural networks.

## REPRODUCIBILITY STATEMENT

In the experiments, we only used publicly available data so that all the results to be reproducible. The code is available at `https://github.com/sato9hara/StableDecisionTree`

## ACKNOWLEDGEMENT

SH is supported by JST, PRESTO Grant Number JPMJPR20C8. YY is supported by JST, PRESTO Grant Number JPMJPR192B.

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

## A    ANALYSIS OF STABLEDR

In this section, we show the following performance guarantee for STABLEDR.

**Theorem A.1.** *For a training data $\mathcal{L}$, a depth bound $B$, and a parameter $\epsilon > 0$, let $\omega =$ STABLEDR$(\mathcal{L}, B, \epsilon)$. Then, we have $\mathbf{E}_\omega\left[\mathrm{opt}_{\omega,B}(\mathcal{L})\right] \geq (1 - \epsilon)\mathrm{opt}_B(\mathcal{L})$. Moreover, we have $\sum_{i=1}^n d_{\mathrm{TV}}(\omega, \omega^{(i)}) = O\left(\frac{\log |\Omega|}{\epsilon}\right)$, where $\omega^{(i)} = A(\mathcal{L}^{(i)}, B, \epsilon)$, and $d_{\mathrm{TV}}(\omega, \omega^{(i)})$ denotes the total variation distance between (the distributions of) $\omega$ and $\omega^{(i)}$.*

The first inequality claims that we can achieve a nearly optimal total score using the output decision rule $\omega$. The second inequality claims that the distribution of $\omega$ does not change significantly when a data point is removed from the training data. Theorem A.1 is obtained by combining Lemmas A.2 and A.4.

### A.1    APPROXIMATION GUARANTEE

First, we show that the selected decision rule does not much deteriorate the total score of an optimal decision tree.

**Lemma A.2.** *Let $\omega =$ STABLEDR$(\mathcal{L}, B, \epsilon)$. Then, we have*

$$\mathbf{E}_\omega[\mathrm{opt}_{\omega,B}(\mathcal{L})] \geq (1 - \epsilon)\mathrm{opt}_B(\mathcal{L}).$$

*Proof.* For any $c > 0$, we have

$$\Pr[\mathrm{opt}_{\omega,B}(\mathcal{L}) \leq \mathrm{opt}_B(\mathcal{L}) - c]$$
$$= \frac{\sum_{\psi \in \Omega : \mathrm{opt}_{\psi,B}(\mathcal{L}) \leq \mathrm{opt}_B(\mathcal{L}) - c} \exp(\lambda \cdot \mathrm{opt}_{\psi,B}(\mathcal{L}))}{\sum_{\psi \in \Omega} \exp(\lambda \cdot \mathrm{opt}_{\psi,B}(\mathcal{L}))}$$
$$\leq \frac{|\Omega| \exp(\lambda \cdot (\mathrm{opt}_B(\mathcal{L}) - c))}{\sum_{\psi \in \Omega} \exp(\lambda \cdot \mathrm{opt}_{\psi,B}(\mathcal{L}))}$$
$$\leq \frac{|\Omega| \exp(\lambda \cdot (\mathrm{opt}_B(\mathcal{L}) - c))}{\exp(\lambda \cdot \mathrm{opt}_B(\mathcal{L}))} \leq |\Omega| \exp(-\lambda c).$$

Therefore, we have

$$\mathbf{E}[\mathrm{opt}_{\omega,B}(\mathcal{L})]$$
$$\geq \Pr[\mathrm{opt}_{\omega,B}(\mathcal{L}) \leq \mathrm{opt}_B(\mathcal{L}) - c] \cdot 0$$
$$\quad + \Pr[\mathrm{opt}_{\omega,B}(\mathcal{L}) > \mathrm{opt}_B(\mathcal{L}) - c] \cdot (\mathrm{opt}_B(\mathcal{L}) - c)$$
$$\geq (1 - |\Omega| \exp(-\lambda c)) \cdot (\mathrm{opt}_B(\mathcal{L}) - c)$$
$$\geq \mathrm{opt}_B(\mathcal{L}) - |\Omega| \exp(-\lambda c) - c.$$

By setting $c = \log |\Omega| / \lambda$ and the choice of $\lambda$, the claim holds. $\square$

### A.2    AVERAGE SENSITIVITY

Next, we analyze the average sensitivity of STABLEDR. For notational simplicity, we write $\mathrm{opt}_\omega$ and $\mathrm{opt}_\omega^{(i)}$ to denote $\mathrm{opt}_{\omega,B}(\mathcal{L})$ and $\mathrm{opt}_{\omega,B}(\mathcal{L}^{(i)})$, respectively. The following lemma is useful for our analysis.

**Lemma A.3.** *For any decision rule $\omega \in \Omega$, we have*

$$\sum_{i=1}^n \left(\mathrm{opt}_\omega - \mathrm{opt}_\omega^{(i)}\right) \leq \mathrm{opt}_\omega.$$

*Similarly, we have*

$$\sum_{i=1}^n \left(\mathrm{opt}_B(\mathcal{L}) - \mathrm{opt}_B(\mathcal{L}^{(i)})\right) \leq \mathrm{opt}_B(\mathcal{L}).$$

*Proof.* We first consider the first statement. Let $\phi$ be the optimal decision that attains $\text{opt}_\omega$. Note that $\phi$ has depth $B$ and the root node of $\phi$ has the decision rule $\omega$. Then, we have

$$\sum_{i=1}^{n}\left(\text{opt}_\omega - \text{opt}_\omega^{(i)}\right) \leq \sum_{i=1}^{n}(s(\phi, \mathcal{L}) - s(\phi, \mathcal{L}^{(i)})) = \sum_{i=1}^{n} 1[\phi(x_i) = y]$$
$$= s(\phi, \mathcal{L}) = \text{opt}_\omega.$$

The second statement follows by a similar argument. □

**Lemma A.4.** *Let*

$$\omega = \text{STABLEDR}(\mathcal{L}, B, \epsilon),$$
$$\omega^{(i)} = \text{STABLEDR}(\mathcal{L}^{(i)}, B, \epsilon).$$

*Then, we have*

$$\sum_{i=1}^{n} d_{\text{TV}}(\omega, \omega^{(i)}) = O\left(\frac{\log |\Omega|}{\epsilon}\right).$$

*Proof.* Notice that

$$\sum_{i=1}^{n} d_{\text{TV}}(\omega, \omega^{(i)}) = \sum_{i=1}^{n} \sum_{\psi \in \Omega} \max\left\{0, \Pr[\omega = \psi] - \Pr[\omega^{(i)} = \psi]\right\}.$$

Let $\lambda^{(i)}$ be $\lambda$ used in $\text{STABLEDR}(\mathcal{L}^{(i)}, B, \epsilon)$. Then we have

$$\max\left\{0, \Pr[\omega = \psi] - \Pr[\omega^{(i)} = \psi]\right\}$$
$$= \max\left\{0, \frac{\exp(\lambda \cdot \text{opt}_\psi)}{\sum_{\psi' \in \Omega} \exp(\lambda \cdot \text{opt}_{\psi'})} - \frac{\exp(\lambda^{(i)} \cdot \text{opt}_\psi^{(i)})}{\sum_{\psi' \in \Omega} \exp(\lambda^{(i)} \cdot \text{opt}_{\psi'}^{(i)})}\right\}$$
$$\leq \frac{\exp(\lambda \cdot \text{opt}_\psi) - \exp(\lambda \cdot \text{opt}_\psi^{(i)})}{\sum_{\psi' \in \Omega} \exp(\lambda \cdot \text{opt}_{\psi'})}$$
$$+ \max\left\{0, \frac{\exp(\lambda \cdot \text{opt}_\psi^{(i)})}{\sum_{\psi' \in \Omega} \exp(\lambda \cdot \text{opt}_{\psi'})} - \frac{\exp(\lambda^{(i)} \cdot \text{opt}_\psi^{(i)})}{\sum_{\psi' \in \Omega} \exp(\lambda^{(i)} \cdot \text{opt}_{\psi'}^{(i)})}\right\}, \tag{3}$$

where the equality is from the design of the algorithm and the inequality is from the following inequality

$$\max\{0, b - a\} \leq (b - x) + \max\{0, x - a\}$$

which holds for any $x \leq b$.

Let $A_{i,\psi}$ and $B_{i,\psi}$ denote the first and second terms, respectively, of (3). The following two claims bound the sums of the first and the second terms over $i$ and $\psi$.

**Claim A.5.**
$$\sum_{i=1}^{n} \sum_{\psi \in \Omega} A_{i,\psi} \leq \lambda \cdot \text{opt}_B(\mathcal{L}).$$

**Claim A.6.**
$$\sum_{i=1}^{n} \sum_{\psi \in \Omega} B_{i,\psi} \leq O(\lambda \cdot \text{opt}_B(\mathcal{L})).$$

Before proving these claims, we first complete the proof of the lemma assuming them. Combining (3) and the two claims above, we have

$$\sum_{i=1}^{n} d_{\text{TV}}(\omega, \omega^{(i)}) \leq O(\lambda \cdot \text{opt}_B(\mathcal{L})) + 1 = O\left(\frac{\log |\Omega|}{\epsilon}\right). \qquad □$$

Theorem A.1 follows by combining Lemmas A.2 and A.4.

*Proof of Claim A.5.* We have

$$\frac{\exp(\lambda \cdot \mathrm{opt}_\psi) - \exp(\lambda \cdot \mathrm{opt}_\psi^{(i)})}{\sum_{\psi' \in \Omega} \exp(\lambda \cdot \mathrm{opt}_{\psi'})} = \Pr[\omega = \psi]\left(1 - \frac{\exp(\lambda \cdot \mathrm{opt}_\psi^{(i)})}{\exp(\lambda \cdot \mathrm{opt}_\psi)}\right)$$

$$= \Pr[\omega = \psi]\left(1 - \exp\left(-\lambda \cdot (\mathrm{opt}_\psi - \mathrm{opt}_\psi^{(i)})\right)\right)$$

$$\leq \lambda \cdot \Pr[\omega = \psi](\mathrm{opt}_\psi - \mathrm{opt}_\psi^{(i)}),$$

where the inequality is from $1 - e^{-x} \leq x$ for any $x \in \mathbb{R}$. Therefore, we have

$$\sum_{i=1}^{n} \sum_{\psi \in \Omega} A_{i,\psi}$$

$$= \sum_{i=1}^{n} \sum_{\psi \in \Omega} \frac{\exp(\lambda \cdot \mathrm{opt}_\psi) - \exp(\lambda \cdot \mathrm{opt}_\psi^{(i)})}{\sum_{\psi' \in \Omega} \exp(\lambda \cdot \mathrm{opt}_{\psi'})}$$

$$\leq \lambda \sum_{i=1}^{n} \sum_{\psi \in \Omega} \Pr[\omega = \psi](\mathrm{opt}_\psi - \mathrm{opt}_\psi^{(i)})$$

$$\leq \lambda \sum_{\psi \in \Omega} \Pr[\omega = \psi]\mathrm{opt}_\psi \qquad \text{(by Lemma A.3)}$$

$$\leq \lambda \cdot \mathrm{opt}_B(\mathcal{L}).$$

as desired. $\qquad\square$

*Proof of Claim A.6.* We first note that

$$B_{i,\psi}$$

$$= \max\left\{0, \frac{\exp(\lambda \cdot \mathrm{opt}_\psi^{(i)})}{\sum_{\psi' \in \Omega} \exp(\lambda \cdot \mathrm{opt}_{\psi'})} - \frac{\exp(\lambda^{(i)} \cdot \mathrm{opt}_\psi^{(i)})}{\sum_{\psi' \in \Omega} \exp(\lambda^{(i)} \cdot \mathrm{opt}_{\psi'}^{(i)})}\right\}$$

$$\leq \max\left\{0, \frac{\exp(\lambda \cdot \mathrm{opt}_\psi^{(i)})}{\sum_{\psi' \in \Omega} \exp(\lambda \cdot \mathrm{opt}_{\psi'})} - \frac{\exp(\lambda^{(i)} \cdot \mathrm{opt}_\psi^{(i)})}{\sum_{\psi' \in \Omega} \exp(\lambda \cdot \mathrm{opt}_{\psi'})}\right\}$$

$$= \max\left\{0, \frac{\exp(\lambda \cdot \mathrm{opt}_\psi^{(i)})}{\sum_{\psi' \in \Omega} \exp(\lambda \cdot \mathrm{opt}_{\psi'})}\left(1 - \exp\left(-\mathrm{opt}_\psi^{(i)}(\lambda - \lambda^{(i)})\right)\right)\right\}$$

$$\leq \max\left\{0, \Pr[\omega = \psi]\mathrm{opt}_\psi^{(i)}(\lambda - \lambda^{(i)})\right\}$$

$$\leq \mathrm{opt}_B(\mathcal{L}) \Pr[\omega = \psi]|\lambda - \lambda^{(i)}|.$$

Also, we have

$$\sum_{i=1}^{n} \left|\lambda - \lambda^{(i)}\right| \leq \lambda \sum_{i=1}^{n} \left|\frac{\frac{\log|\Omega|}{\mathrm{opt}_B(\mathcal{L})} - \frac{\log|\Omega|}{\mathrm{opt}_B(\mathcal{L}^{(i)})}}{\frac{\log|\Omega|}{\mathrm{opt}_B(\mathcal{L})}}\right|$$

$$\leq \lambda \sum_{i=1}^{n} \max\left\{\frac{\frac{\log|\Omega|}{\mathrm{opt}_B(\mathcal{L})} - \frac{\log|\Omega|}{\mathrm{opt}_B(\mathcal{L}^{(i)})}}{\frac{\log|\Omega|}{\mathrm{opt}_B(\mathcal{L})}}, \frac{\frac{\log|\Omega|}{\mathrm{opt}_B(\mathcal{L}^{(i)})} - \frac{\log|\Omega|}{\mathrm{opt}_B(\mathcal{L})}}{\frac{\log|\Omega|}{\mathrm{opt}_B(\mathcal{L})}}\right\}$$

$$\leq \lambda \sum_{i=1}^{n} \max\left\{\frac{\log|\Omega| - \log|\Omega|}{\log|\Omega|}, \frac{\frac{1}{\mathrm{opt}_B(\mathcal{L}^{(i)})} - \frac{1}{\mathrm{opt}_B(\mathcal{L})}}{\frac{1}{\mathrm{opt}_B(\mathcal{L})}}\right\}$$

$$\leq \lambda \sum_{i=1}^{n} \frac{\log|\Omega| - \log|\Omega|}{\log|\Omega|} + \lambda \sum_{i=1}^{n} \frac{\mathrm{opt}_B(\mathcal{L}) - \mathrm{opt}_B(\mathcal{L}^{(i)})}{\mathrm{opt}_B(\mathcal{L}^{(i)})}$$

$$\leq \lambda \sum_{i=1}^{n} \frac{\log |\Omega| - \log |\Omega|}{\log |\Omega|} + 2\lambda \sum_{i=1}^{n} \frac{\mathrm{opt}_B(\mathcal{L}) - \mathrm{opt}_B(\mathcal{L}^{(i)})}{\mathrm{opt}_B(\mathcal{L})} \qquad \text{(by } \mathrm{opt}_B(\mathcal{L}^{(i)}) \geq 1\text{)}$$

$$= O(\lambda), \qquad\qquad\qquad\qquad\qquad\qquad\qquad\qquad \text{(by Lemma A.3)}$$

Combining the two inequalities above, we obtain

$$\sum_{i=1}^{n} \sum_{\psi \in \Omega} B_{i,\psi} \leq \sum_{i=1}^{n} \sum_{\psi \in \Omega} \mathrm{opt}_B \Pr[\omega = \psi] |\lambda - \lambda^{(i)}| = O\left(\lambda \cdot \mathrm{opt}_B(\mathcal{L})\right). \qquad \square$$

## B   ANALYSIS OF STABLEDT

In this section, we analyze STABLEDT and prove Theorem 4.1

*Proof of the first claim of Theorem 4.1.* Let $\mathcal{L}_0$ be the input training data (so that we can use $\mathcal{L}$ to denote other sets).

We prove the following by backward induction on depth.

$$\mathbf{E}[s(\text{STABLEDT'}(\mathcal{L}, B, \epsilon, d), \mathcal{L})] \geq (1 - \epsilon)^{B-d} \mathrm{opt}_B(\mathcal{L}).$$

Then, the statement holds by setting $d = 0$.

The claim clearly holds when $d = B$ because we output the optimal label.

Suppose that the claim holds for depth more than $d$. Consider a particular call STABLEDT'$(\mathcal{L}, B, \epsilon, d)$, and let $\phi$ denote the output decision tree, let $\omega$ be the decision rule used in the root node of $\phi$, and let $\mathcal{L}_L^\omega$ and $\mathcal{L}_R^\omega$ denote the two training datas obtained from $\mathcal{L}$ by splitting it according to $\omega$. Note that these are random variables. Then, we have

$$\mathbf{E}_\phi[s(\phi, \mathcal{L})] = \sum_{\psi \in \Omega} \Pr[\omega = \psi] \Big( \mathbf{E}[s(\text{STABLEDT'}(\mathcal{L}_L^\psi, B, \epsilon, d+1), \mathcal{L}_L^\psi)]$$

$$+ \mathbf{E}[s(\text{STABLEDT'}(\mathcal{L}_R^\psi, B, \epsilon, d+1), \mathcal{L}_R^\psi)] \Big)$$

$$\geq \sum_{\psi \in \Omega} \Pr[\omega = \psi] \Big( (1 - \epsilon)^{B-d-1} \mathrm{opt}_{B-d-1}(\mathcal{L}_L^\psi)$$

$$+ (1 - \epsilon)^{B-d-1} \mathrm{opt}_{B-d-1}(\mathcal{L}_R^\psi) \Big)$$

$$\geq (1 - \epsilon)^{B-d-1} \sum_{\psi \in \Omega} \Pr[\omega = \psi] \Big( \mathrm{opt}_{\psi, B-d}(\mathcal{L}) \Big)$$

$$\geq (1 - \epsilon)^{B-d} \sum_{\psi \in \Omega} \Pr[\omega = \psi](1 - \epsilon) \mathrm{opt}_{B-d}(\mathcal{L})$$

$$\geq (1 - \epsilon)^{B-d} \mathrm{opt}_{B-d}(\mathcal{L}),$$

where the first inequality is based on the induction hypothesis and the second to last inequality is based on Theorem A.1. $\qquad \square$

*Proof of the second claim of Theorem 4.1.* For notational simplicity, we drop the arguments $B$ and $\epsilon$ when calling STABLEDT'$(\mathcal{L}, B, \epsilon, d)$, because they are fixed in this proof. Additionally, we write STABLEDT' instead of STABLEDT'.

Let $\mathcal{L}_0 = ((x_1, y_1), \ldots, (x_n, y_n))$ be the input training data (so that we can use $\mathcal{L}$ to denote other sets). For a subset $\mathcal{L}$ of $\mathcal{L}_0$ and $i \in \{1, 2, \ldots, n\}$, let $\mathcal{L}^{(i)} := \mathcal{L} \setminus \{(x_i, y_i)\}$.

For $0 \leq d \leq B$, let $\mathcal{L}_{d,1}, \ldots, \mathcal{L}_{d,2^d}$ be the sets on which STABLEDT' is called at depth $d$ (if the number of sets on which STABLEDT' is called at depth $d$ is less than $2^d$, we append empty sets). We can order them so that STABLEDT'$(\mathcal{L}_{d,j}, d)$ calls STABLEDT'$(\mathcal{L}_{d+1,2j-1}, d+1)$ and STABLEDT'$(\mathcal{L}_{d+1,2j}, d+1)$ (if STABLEDT'$(\mathcal{L}_{d,j}, d)$ does not make recursive calls, we set $\mathcal{L}_{d+1,2j-1} = \mathcal{L}_{d+1,2j} = \emptyset$).

For fixed $\{\mathcal{L}_{B,j}\}_j$, we have

$$\sum_{i=1}^{n} \sum_{j=1}^{2^d} d_{\mathrm{EM}}(\mathrm{STABLEDT'}(\mathcal{L}_{B,j}, B), \mathrm{STABLEDT'}(\mathcal{L}_{B,j}^{(i)}, B))$$

$$\leq \sum_{i=1}^{n} \sum_{j=1}^{2^d} \mathbb{1}[(x_i, y_i) \in \mathcal{L}_{B,j}] = \sum_{i=1}^{n} \mathbb{1}[(x_i, y_i) \in \mathcal{L}] = |\mathcal{L}|$$

because the output changes only when $(x_i, y_i) \in \mathcal{L}$ and the output change is bounded by one.

Let $0 \leq d < B$. Let $\omega_{d,j}$ and $\omega_{d,j}^{(i)}$ be the $\omega$ values used in $\mathrm{STABLEDT'}(\mathcal{L}_{d,j}, d)$ and $\mathrm{STA\text{-}BLEDT'}(\mathcal{L}_{d,j}^{(i)}, d)$, respectively. Note that they are random variables. For a rule $\omega$, Let $\mathcal{L}_{d+1,2j-1}^{\omega}$ and $\mathcal{L}_{d+1,2j}^{\omega}$ be the two sets obtained by partitioning $\mathcal{L}_{d,j}$ according to $\omega$. Then for fixed $\{\mathcal{L}_{d,j}\}_j$, we have

$$\sum_{i=1}^{n} \sum_{j=1}^{2^d} d_{\mathrm{EM}}(\mathrm{STABLEDT'}(\mathcal{L}_{d,j}, d), \mathrm{STABLEDT'}(\mathcal{L}_{d,j}^{(i)}, d))$$

$$\leq \sum_{i=1}^{n} \sum_{j=1}^{2^d} \Big( d_{\mathrm{TV}}(\omega_{d,j}, \omega_{d,j}^{(i)}) \cdot 2^{B-d}$$

$$+ \mathop{\mathbf{E}}_{\omega_{d,j}} d_{\mathrm{EM}}(\mathrm{STABLEDT'}(\mathcal{L}_{d+1,2j-1}^{\omega_{d,j}}, d+1), \mathrm{STABLEDT'}(\mathcal{L}_{d+1,2j-1}^{\omega_{d,j},(i)}, d+1))$$

$$+ \mathop{\mathbf{E}}_{\omega_{d,j}} d_{\mathrm{EM}}(\mathrm{STABLEDT'}(\mathcal{L}_{d+1,2j}^{\omega_{d,j}}, d+1), \mathrm{STABLEDT'}(\mathcal{L}_{d+1,2j}^{\omega_{d,j},(i)}, d+1)) \Big)$$

$$\leq C \cdot 2^{B-d} \sum_{j=1}^{2^d} \frac{\log|\Omega|}{\epsilon}$$

$$+ \sum_{i=1}^{n} \sum_{j=1}^{2^{d+1}} \mathop{\mathbf{E}}_{\omega_{d,j}} d_{\mathrm{EM}}(\mathrm{STABLEDT'}(\mathcal{L}_{d+1,j}^{\omega_{d,j}}, d+1), \mathrm{STABLEDT'}(\mathcal{L}_{d+1,j}^{\omega_{d,j},(i)}, d+1))$$

$$\text{(by Lemma A.1)}$$

$$\leq C \cdot 2^{B} \frac{\log|\Omega|}{\epsilon}$$

$$+ \sum_{i=1}^{n} \sum_{j=1}^{2^{d+1}} \mathop{\mathbf{E}}_{\omega_{d,j}} d_{\mathrm{EM}}(\mathrm{STABLEDT'}(\mathcal{L}_{d+1,j}^{\omega_{d,j}}, d+1), \mathrm{STABLEDT'}(\mathcal{L}_{d+1,j}^{\omega_{d,j},(i)}, d+1)),$$

where $C > 0$ is some universal constant. By backward induction, we obtain for any $d$ and fixed $\{\mathcal{L}_{d,j}\}_j$

$$\sum_{i=1}^{n} \sum_{j=1}^{2^d} d_{\mathrm{EM}}(\mathrm{STABLEDT'}(\mathcal{L}_{d,j}, d), \mathrm{STABLEDT'}(\mathcal{L}_{d,j}^{(i)}, d))$$

$$\leq C \cdot 2^{B}(B-d) \frac{\log|\Omega|}{\epsilon} + n$$

for every $0 \leq d \leq B$. By setting $d = 0$, we obtain the claim. $\qquad\square$

## C  MISSING PROOFS OF SECTION 5

In this section, we prove Theorem 5.1. We discuss modifications to STABLEDR and STABLEDT in Sections C.1 and C.2, respectively.

---

**Algorithm 4:**

---

1 **Procedure** SEEDEDSTABLEDR($\mathcal{L}, B, \epsilon, \pi$)

2 $\quad$ $\lambda \leftarrow \frac{2 \log |\Omega|}{\epsilon \cdot \mathrm{opt}_B(\mathcal{L})}$;

3 $\quad$ **while true do**

4 $\quad\quad$ Sample $\omega \in \Omega$ uniformly at random using $\pi$;

5 $\quad\quad$ Sample $\tau \in [0, 1]$ uniformly at random using $\pi$;

6 $\quad\quad$ $p_\omega$ be the probability of choosing $\omega$ as given in STABLEDR;

7 $\quad\quad$ **if** $p_\omega > \tau$ **then return** $\omega$;

8 **Procedure** SEEDEDSTABLEDT'($\mathcal{L}, B, \epsilon, d, j, \pi$)

9 $\quad$ **if** $|\mathcal{L}| \leq 1$ *or* $d = B$ **then**

10 $\quad\quad$ **return** an optimal label for $\mathcal{L}$.

11 $\quad$ $\omega \leftarrow$ SEEDEDSTABLEDR($\mathcal{L}, B, \epsilon, \pi$);

12 $\quad$ Partition $\mathcal{L}$ into $\mathcal{L}_L \cup \mathcal{L}_R$ according to $\omega$;

13 $\quad$ $\pi_L \leftarrow (\pi_1, \pi_3, \ldots)$ and $\pi_R \leftarrow (\pi_2, \pi_4, \ldots)$;

14 $\quad$ $\phi_L \leftarrow$ SEEDEDSTABLEDT'($\mathcal{L}_L, B, \epsilon, d+1, 2j, \pi_L$);

15 $\quad$ $\phi_R \leftarrow$ SEEDEDSTABLEDT'($\mathcal{L}_R, B, \epsilon, d+1, 2j+1, \pi_R$);

16 $\quad$ Let $\phi_\omega$ be the decision tree such that the root node $t$ has rule $\omega$ and the left and right
$\quad\quad$ children of $t$ are $\phi_L$ and $\phi_R$, respectively;

17 $\quad$ **return** $\phi_\omega$.

18 **Procedure** SEEDEDSTABLEDT($\mathcal{L}, B, \epsilon, \pi$)

19 $\quad$ **return** SEEDEDSTABLEDT'($\mathcal{L}, B, \epsilon, 0, 1, \pi$).

---

### C.1 DECISION RULE SELECTION

In STABLEDR, we sampled a rule $\omega \in \Omega$ by the exponential mechanism McSherry & Talwar (2007). To bound the expected deterministic average sensitivity over random bits, we perform the following rejection sampling. We first sample a rule $\omega \in \Omega$ and threshold $\tau \in [0, 1]$ uniformly at random by using $\pi$. If the threshold $\tau$ is more than the probability $p_\omega$ that we sample $\omega$ in the exponential mechanism, then we output $\omega$. Otherwise, we repeat the same process again. The details are given as SEEDEDSTABLEDR in Algorithm 4.

The following lemma shows that the distributions of STABLEDR and DERANDOMIZEDSTABLEDR are the same and the derandomized average sensitivity of the latter can be bounded from above by the average sensitivity of the former.

**Lemma C.1.** *Let*

$$
\begin{aligned}
\omega &= \text{STABLEDR'}(\mathcal{L}, B, \epsilon), \\
\omega^{(i)} &= \text{STABLEDR'}(\mathcal{L}^{(i)}, B, \epsilon), \\
\omega_\pi &= \text{DERANDOMIZEDSTABLEDR'}(\mathcal{L}, B, \epsilon, \pi), \\
\omega_\pi^{(i)} &= \text{DERANDOMIZEDSTABLEDR'}(\mathcal{L}^{(i)}, B, \epsilon, \pi).
\end{aligned}
$$

*Then, the distribution of $\omega$ and that of $\omega_\pi$ over $\pi$ are the same. Moreover for any $i \in \{1, 2, \ldots, n\}$, we have*

$$
\Pr_\pi[\omega_\pi \neq \omega_\pi^{(i)}] \leq 2d_{\mathrm{TV}}(\omega, \omega^{(i)}).
$$

*Proof.* The first claim is clear from the design of DERANDOMIZEDSTABLEDR.

Now we see the second claim. Let $Z = \sum_{\psi \in \Omega} \exp(\lambda \cdot \mathrm{opt}_{\psi,B}(\mathcal{L}))$ and let $Z^{(i)} = \sum_{\psi \in \Omega} \exp(\lambda \cdot \mathrm{opt}_{\psi,B}(\mathcal{L}^{(i)}))$. For $\omega \in \Omega$, we let $p_\omega = \exp(\lambda \cdot \mathrm{opt}_{\omega,B}(\mathcal{L}))/Z$. For $\omega \in \Omega$, we let $p_\omega^{(i)} = \exp(\lambda \cdot \mathrm{opt}_{\omega,B}(\mathcal{L}^{(i)}))/Z^{(i)}$, and for $\omega \in \Omega \setminus \Omega$, we let $p_\omega^{(i)} = 0$. Then, we have

$$
\Pr_\pi[\omega_\pi \neq \omega_\pi^{(i)}] \leq \sum_{\psi \in \Omega} \Pr_\tau[\min\{p_\psi, p_\psi^{(i)}\} < \tau < \max\{p_\psi, p_\psi^{(i)}\}]
$$

$$= \sum_{\psi \in \Omega} |p_\psi - p_\psi^{(i)}| = 2d_{\mathrm{TV}}(\omega, \omega^{(i)}). \qquad \square$$

By the analysis of STABLEDR and Lemma C.1, we obtain the following:

**Theorem C.2.** *Let* $\omega_\pi = \text{SEEDEDSTABLEDR}(\mathcal{L}, B, \epsilon, \pi)$. *We have* $\mathbf{E}_\pi[\text{opt}_{\omega_\pi, B}(\mathcal{L})] \geq (1 - \epsilon)\text{opt}_B(\mathcal{L})$. *Moreover for* $\omega_\pi^{(i)} = \text{SEEDEDSTABLEDR}(\mathcal{L}^{(i)}, B, \epsilon, \pi)$, *we have* $\mathbf{E}_\pi \left[ \sum_{i=1}^n d_{\mathrm{DT}}(\omega_\pi, \omega_\pi^{(i)}) \right] = O\left( \frac{\log |\Omega|}{\epsilon} \right)$.

## C.2 DECISION TREE CONSTRUCTION

We now explain the modification to STABLEDT. Let $\mathcal{L}_{d,1}, \ldots, \mathcal{L}_{d,2^d}$ be the sets on which our algorithm is called at depth $d$ as defined in the proof in Section B. Then, we want to make sure that the same random bits are used when processing particular $\mathcal{L}_{d,j}$ no matter whether the input training data is $\mathcal{L}$ or $\mathcal{L}^{(i)}$ $(1 \leq i \leq n)$. To this end, at each node in the decision tree, we split the random bits $\pi = (\pi_1, \pi_2, \ldots)$ into $\pi_L = (\pi_1, \pi_3, \ldots)$ and $\pi_R = (\pi_2, \pi_4, \ldots)$, and then pass $\pi_L$ and $\pi_R$ on to the nodes for $\mathcal{L}_{d+1,2j}$ and $\mathcal{L}_{d+1,2j+1}$, respectively. See Algorithm 4 for details.

We replace Theorem A.1 with Theorem C.2 in the proof of Theorem 4.1, and we obtain Theorem 5.1.

# D ADDITIONAL RESULTS

## D.1 DETAILED RESULTS IN SECTION 7.2

In Section 7.2, we reported the trends of average sensitivity and accuracies over $\epsilon$ on small datasets, breast cancer and diabetes. Here, we show the detailed results (i) with error bars, and (ii) with a relaxed version of the tree distance. For (i), in addition to the average results, we also show their variations. More specifically, we report the 25 and 75 percentiles of the results over 10 random realizations of the sampled training data. For (ii), we adopt a relaxed version of the tree distance in Algorithm 5. In the original tree distance in Algorithm 2, we regarded that two trees are (completely) different when their top rules are different (Line 6). In the relaxed version in Algorithm 5, we regard that two trees are completely different only when the features used in the top rules are different. With this relaxation, we regard two subtrees with similar top rules such as $\omega : u \mapsto 1[u_1 \leq 1.0]$ and $\omega' : u \mapsto 1[u_1 \leq 1.01]$ as identical.

---

**Algorithm 5:**

1 **Procedure** DISTANCE'$(\phi, \phi')$
2      Let $t$ and $t'$ be the root nodes of $\phi$ and $\phi'$, respectively;
3      Let $\omega$.feature be the feature used in $\omega$;
4      **if** *both $t$ and $t'$ are leaves* **then**
5          **return** 0 if $y_t = y_{t'}$ and 2 otherwise.
6      **else if** *either $t$ or $t'$ is a leaf* **then return** $|\phi| + |\phi'|$;
7      **else if** $\omega_t$.feature $\neq \omega_{t'}$.feature **then return** $|\phi| + |\phi'|$;
8      **else**
9          Let $\phi_L, \phi_R$ be the decision trees rooted at the left and right children of $t$, respectively;
10          Let $\phi'_L, \phi'_R$ be the decision trees rooted at the left and right children of $t'$, respectively;
11          **return** DISTANCE'$(\phi_L, \phi'_L)$ + DISTANCE'$(\phi_R, \phi'_R)$.

---

Figures 5 and 6 show the detailed results on breast cancer and diabetes, respectively. In the figures, we show the 25 and 75 percentiles using colored shades. The figures named Sensitivity and Sensitivity' are the average sensitivity computed using the original distance and the relaxed distance, respectively.

The figures confirm that the decrease of the average sensitivity for $\epsilon > 0.1$ will be sufficiently significant, in particular for the number of data removal $m = 1\%$ and $10\%$. The figures on Sensitivity and Sensitivity' also confirm that the average sensitivity measured by using the original tree distance

and the relaxed tree distance are almost identical, implying the choice of the tree distance will only have negligible impacts to the results.

Figure 5: Detailed results on average sensitivity and accuracy of the trained trees over different $\epsilon$ on breast cancer. The figures named Sensitivity and Sensitivity' are the average sensitivity computed using the original distance and the relaxed distance, respectively.

## D.2 TEST ACCURACY

For the experiments in Section 7, Figure 7 shows the trade-off curves between average sensitivity and test accuracy when $\epsilon$ is changed.

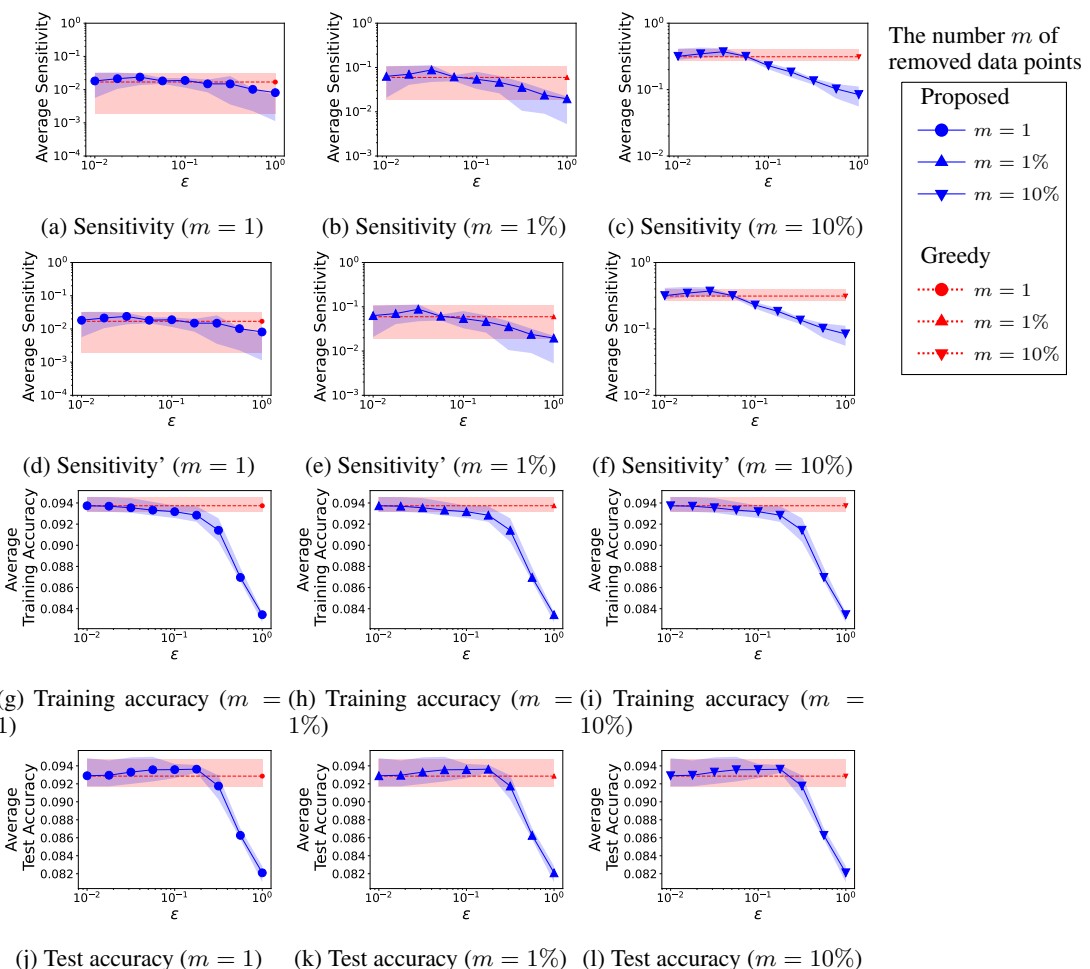

Figure 6: Detailed results on average sensitivity and accuracy of the trained trees over different $\epsilon$ on diabetes. The figures named Sensitivity and Sensitivity' are the average sensitivity computed using the original distance and the relaxed distance, respectively.

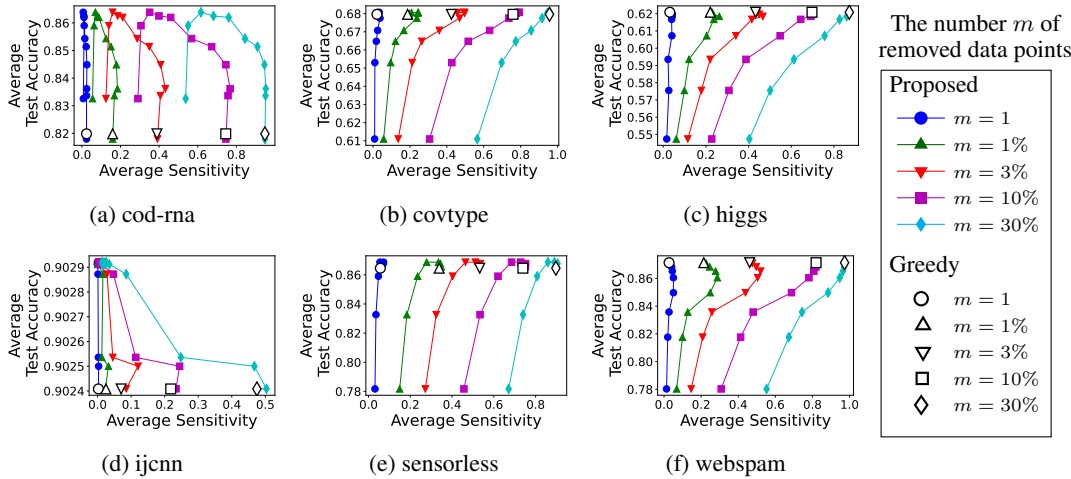

Figure 7: Trade-off curves between average sensitivity and test accuracy when $\epsilon$ is changed. We varied the number of training data points to be removed from one to 30% of the sampled training data. White markers denote the results for the greedy tree learning.

