# OpenReview forum: "Average Sensitivity of Decision Tree Learning"
_ICLR.cc/2023/Conference — ICLR 2023 poster_

### Official Review · Reviewer_sr4W · 2022-10-25

**Confidence:** 3
**Correctness:** 3
**Technical Novelty And Significance:** 3
**Empirical Novelty And Significance:** 3
**Recommendation:** 6

**Clarity, Quality, Novelty And Reproducibility:**

The paper is presented very clearly, and the theory and experiments exhibit high quality work. The use of randomized selection rules is very common in statistical literature on tree-based models. They occur in random forests, and extremely randomized trees (Guerts et al. 2006). I believe a connection could be drawn, since randomization is used in those cases to stabilize the resulting ensembled model (in the statistical sense, i.e. variance reduction).

**Strength And Weaknesses:**

Strengths
Clear, well-written, principled approach to an important problem. The theoretical results clearly and cleanly demonstrate the performance tradeoffs. The analysis provides stronger bounds than those available from standard DP technique.

Weaknesses
The claim in the abstract - “The experimental results on real-world datasets demonstrate that the proposed algorithm achieves a low average sensitivity with an insignificant decrease in accuracy” - seems not justified. If I understand figure 4 correctly, there is in fact a significant decrease in accuracy for low sensitivity approaches, on many of the datasets.


**Summary Of The Paper:**

The authors propose a method to stabilize decision tree learning by randomizing the decision rule for selecting the split. The authors prove several theorems that quantify the accuracy / stability tradeoff, and perform experiments on several datasets of varying sizes to demonstrate the tradeoff in practice.

**Summary Of The Review:**

A simple modification to standard tree-building procedures provides some nice guarantees about model stability with respect to ablation of small amounts of training data.  The paper is well-written and technically sound.  The main contribution seems to be the theoretical analysis that precisely quantifies the accuracy and sensitivity bounds. The experimental results do show a tradeoff occurs in practice, somewhat in contradiction to the abstract.

---

> ### Author Response · Authors · 2022-11-08
> **Reply to the review**
>
> We thank the reviewer for providing helpful comments.
>
>
> > The claim in the abstract - “The experimental results on real-world datasets demonstrate that the proposed algorithm achieves a low average sensitivity with an insignificant decrease in accuracy” - seems not justified. If I understand figure 4 correctly, there is in fact a significant decrease in accuracy for low sensitivity approaches, on many of the datasets.
>
> We cofirm that we need to sacrifice a portion of the accuracy to achieve a low average sensitivity. We revised the statement focusing more on the flexibility of the proposed algorithm about the trade-off between average sensitivity and accuracy.
>
>
> > The use of randomized selection rules is very common in statistical literature on tree-based models. They occur in random forests, and extremely randomized trees (Guerts et al. 2006). I believe a connection could be drawn, since randomization is used in those cases to stabilize the resulting ensembled model (in the statistical sense, i.e. variance reduction).
>
> The raised techniques above are for ensemble trees where the stability (or, variance reduction) is achieved by averaging multiple trees. As noted in the first paragraph of Section 2, we study average sensitivity of algorithms for learning single trees because they are used to understand the structure in the input data and in this context the stability of the output tree is important. For single trees, we cannot use the raised averaging methods, and hence we need to develop a completely different approach. To our knowledge, stabilization of training algorithms without averaging trees is less explored, and we will be the first one who proposed such an algorithm for training single trees.

---

### Official Review · Reviewer_hkCt · 2022-10-25

**Confidence:** 3
**Correctness:** 3
**Technical Novelty And Significance:** 3
**Empirical Novelty And Significance:** 3
**Recommendation:** 6

**Clarity, Quality, Novelty And Reproducibility:**

According to my knowledge the work is novel and original.

Some comments to improve the paper:
- Authors formulate their theory only on the basis of a classification problem. Clearly this is very important, but what about other tasks? I’m thinking to regression, but also clustering, density estimation or anomaly detection… Any idea of how to extend this framework to such cases? For regression it should not be so complicated, whereas for others it can be, especially because we don’t have the concept of “optimal model”.

- Please provide a text explanation of the intuition behind the algorithm proposed in Alg 3.

- Please add around the paper the reference to the material presented in the supplementary part (for example, after presenting the theorems, please add a quick sentence saying that the proof is in the appendix)

- In section 7.3 authors derive some conclusive claims on their methods, but the experiment is based on a single run, with a fixed parametrization of epsilon and m and on a single dataset. Of course this may be interesting for illustrative purposes, but please do not derive conclusive statements on the method.

- In table 1 there is reported the size of the training set, even if only 1000 samples are selected in each run from the second group of datasets (and 80% from the first). Maybe this should be made clearer in the table.



**Strength And Weaknesses:**

Positive points
- The topic is definitely interesting
- The idea, according to my knowledge, is novel and well explained
- The structure of the manuscript is clear
- Experiments and findings are interesting

Negative points
1. I’m not completely convinced about the significance of the proposed ideas. More in detail, I understand that it is interesting to evaluate the sensitivity of tree learning for small changes in the training data, but maybe this is of limited help. Actually Random Forests (or in general ensemble tree methods) start from this diversity and exploit this behavior to enrich single trees, why do we need to correct this? Why do we would prefer to have less sensitive decision trees? Or, in a provocative perspective, why should we work with single trees? I think this is an issue that deserves a discussion.
A second comment: the theorem is focused on error on the training set, which is not as interesting as the error on the testing set (the generalization error). Other analyses focus on the generalization error.


2. I have some concerns on the experiments.
- The choice of the datasets is not well motivated. Moreover, it is not clear why authors used different sets of problems for different parts of the experimental evaluation: in section 7.2 they used breast cancer and diabetes, in section 7.3 breast, and is section 7.4 all the large datasets.
- From table 1 it seems that trees for 3 of the 8 datasets have a very low depth, with two having depth 1 (which, according to me, can not be considered as a tree anymore). Why not using a standard depth, such as log(n)? Comparing results from so differently deep trees may lead to biases in the presented results: did authors consider the tree depth in the analysis?
- Authors repeated the whole procedure 10 times, meaning that the variability across different repetitions can be very high (considering that strong subsampling has been done to create a training set, especially for large datasets). Results only display averaged accuracies, what about standard deviations? Are differences in the plot statistically significant? Without a statistical analysis strong conclusions can not be derived.

3. A minor concern: authors defined “perturbation” as the removal of a single (or more than one) element from the training set. This is not the only possibility, authors should discuss relation with other definitions, like adding noise, or resampling like in bootstrap, just to cite a few.

**Summary Of The Paper:**

This paper presents an analysis of the sensitivity of a learned decision tree when perturbing the learning set. Authors define perturbation and design a training algorithm which is stable with respect to such perturbation, providing a theoretical quantification of it.

**Summary Of The Review:**

Interesting paper with a nice idea. Experiments not completely convincing, some doubts on the significance.

UPDATE AFTER REBUTTAL. I carefully read the responses, the clarifications, and the additional material, and I thank the authors for the significant efforts made in clarifying my doubts. After all, I still maintain my opinion on the paper: this represents an interesting work, with few limitations on experiments and significance (even if I acknowledge the efforts made by the authors in improving these aspects in the rebuttal)

---

> ### Author Response · Authors · 2022-11-08
> **Reply to the review (1/2)**
>
> We thank the reviewer for providing helpful comments. We provide detailed answers below.
>
> > I’m not completely convinced about the significance of the proposed ideas. More in detail, I understand that it is interesting to evaluate the sensitivity of tree learning for small changes in the training data, but maybe this is of limited help. Actually Random Forests (or in general ensemble tree methods) start from this diversity and exploit this behavior to enrich single trees, why do we need to correct this? Why do we would prefer to have less sensitive decision trees? Or, in a provocative perspective, why should we work with single trees? I think this is an issue that deserves a discussion.
>
> As noted in the first paragraph of Section 2, we study the average sensitivity of algorithms for learning single trees because they are used to understand the structure in the input data and in this context the stability of the output tree is important. Ensemble methods mainly focus on accuracy and hence stability of each individual tree is less important.
>
> > A second comment: the theorem is focused on error on the training set, which is not as interesting as the error on the testing set (the generalization error). Other analyses focus on the generalization error.
>
> Any uniform bound (such as the bounds obtained via Rademacher complexity) can be used to bound the generalization error no matter what the output decision tree is. So it suffices to bound the error on the training set in order to bound that on the test set.
>
> > The choice of the datasets is not well motivated. Moreover, it is not clear why authors used different sets of problems for different parts of the experimental evaluation: in section 7.2 they used breast cancer and diabetes, in section 7.3 breast, and is section 7.4 all the large datasets.
>
> We used standard tabular datasets widely used to evaluate machine learning algorithms in many studies. Because stability is particularly crucial for small datasets, we assessed the capability of the proposed algorithm in detail in Sections 7.2 and 7.3 on small datasets. In Section 7.4, we demonstrate that the similar trade-offs we found in small datasets hold also on larger datasets. These are the reasons why we used different datasets across the sections.
>
> > From table 1 it seems that trees for 3 of the 8 datasets have a very low depth, with two having depth 1 (which, according to me, can not be considered as a tree anymore). Why not using a standard depth, such as log(n)? Comparing results from so differently deep trees may lead to biases in the presented results: did authors consider the tree depth in the analysis?
>
> As we remarked in the second paragraph of Section 7.1, we selected the depth by cross-validation. We believe this way of choosing the depth (more generally, hyperparameters) is standard. If we choose the depth to log2(n) as suggested, each leaf node contains exactly one data point if the tree is balanced, which is apparently an overfitting. We presume that the suggested depth log(n) is typically used for tree ensembles rather than single trees.
>
> > Authors repeated the whole procedure 10 times, meaning that the variability across different repetitions can be very high (considering that strong subsampling has been done to create a training set, especially for large datasets). Results only display averaged accuracies, what about standard deviations? Are differences in the plot statistically significant? Without a statistical analysis strong conclusions can not be derived.
>
> Following the suggestion, we added figures with errors in Appendix D.1 in the revised version.
>
> > A minor concern: authors defined “perturbation” as the removal of a single (or more than one) element from the training set. This is not the only possibility, authors should discuss relation with other definitions, like adding noise, or resampling like in bootstrap, just to cite a few.
>
> It is an interesting future direction. An issue in adding noise is that it seems to be hard to obtain stable algorithms when the added noise is an outlier. We do not have this issue in the removal setting because we remove a random data point and hence the probability that we remove an outlier in the input data is small. Because resampling can be seen as the process of removing random data points multiple times, we believe we can obtain a bound on the sensitivity for resampling using the bounds shown in the paper.

---

> ### Author Response · Authors · 2022-11-08
> **Reply to the review (2/2)**
>
> > Authors formulate their theory only on the basis of a classification problem. Clearly this is very important, but what about other tasks? I’m thinking to regression, but also clustering, density estimation or anomaly detection… Any idea of how to extend this framework to such cases? For regression it should not be so complicated, whereas for others it can be, especially because we don’t have the concept of “optimal model”.
>
> Average sensitivity can be considered for any problem if we can define the distance between outputs. For example, [Peng & Yoshida, 2020] studied spectral clustering and [Ito & Yoshida, 2022] studied k-means clustering.
>
> - [Peng & Yoshida, 2020]  Average sensitivity of spectral clustering. KDD 2020.
> - [Ito & Yoshida, 2022] Average Sensitivity of Euclidean k-Clustering. NeurIPS 2022.
>
>
> > Please provide a text explanation of the intuition behind the algorithm proposed in Alg 3.
>
> Following the suggestion, we added an explanation at the end of Section 4 in the revised version.
>
>
> > Please add around the paper the reference to the material presented in the supplementary part (for example, after presenting the theorems, please add a quick sentence saying that the proof is in the appendix)
>
> Following the suggestion, we remarked at the end of the introduction in the revised version.
>
> > In section 7.3 authors derive some conclusive claims on their methods, but the experiment is based on a single run, with a fixed parametrization of epsilon and m and on a single dataset. Of course this may be interesting for illustrative purposes, but please do not derive conclusive statements on the method.
>
> We agree that we cannot conclude anything from the result on a single run. Here, we would like to raise two remarks. First, the essential comparisons are conducted in Section 7.2, showing the trends on average sensitivities as well as accuracies over $\epsilon$. The purpose of Section 7.3 is to investigate in detail how the proposed algorithm attained smaller average sensitivity, by focusing on a single run. Second, we have not concluded anything in Section 7.3. By looking into Section 7.3, we can find that the findings from this single run are referred to as ``implication’’.
>
> > In table 1 there is reported the size of the training set, even if only 1000 samples are selected in each run from the second group of datasets (and 80% from the first). Maybe this should be made clearer in the table.
>
> Following the suggestion, we revised accordingly in the revised version.

---

### Official Review · Reviewer_tBhR · 2022-10-30

**Confidence:** 2
**Correctness:** 3
**Technical Novelty And Significance:** 3
**Empirical Novelty And Significance:** 3
**Recommendation:** 5

**Clarity, Quality, Novelty And Reproducibility:**

Clarity: the paper is well-written, and I think could be quite readable for people familiar with the involved techniques.

Quality: the paper quality is generally fine

Novelty: appear to be quite ok

Reproducibility: I could download the package but not install all dependencies, but there is a clear effort to make the code available for basic testing.

**Strength And Weaknesses:**

+: an interesting problem treated in an original way, that I did not know of

-: some unclarity regarding experiments and methodology that I give here, that authors could maybe clarify:

* If I understand Algorithm 3 well, each potential cut or optimal split of the tree is now a probabilistic decision (the cut is made or not). I honestly do not understand how this result in more stability of the tree? In particular, one would expect randomized cuts to put more diversity in built trees? What is the general value of the probability of accepting a cut (it is hard to grasp intuitively at first sight without being familiar with the approach). For example, in Figure 3.b, how does randomization help to have more systematic obtention of the original tree in the second line?

* It would be good to have some idea of the variance values over some of the graphs. For example, in Figure 2.c, one would expect the high variations to be due to variance and not general trends, especially as those looks very small and edgy?

* In figure 4, it seems that the evolution of all the curves in terms of training accuracy is the same for all the settings (for each curve, the sequence of ordinate values are exactly the same. Given the fact that this are randomized splits over different data sets, how is this possible?

-: some unclarity about the reasonableness of the made assumptions, the considered distance and the obtained results:

* I would disagree to some extent that a distance based on the maximal common sub-tree is a natural distance between learned decision trees. In particular, it seems the distance will consider that a split made in the same variable but on a possibly slightly different value will lead to different trees, which is highly questionable interpretability-wise, as two trees with all the same variables but different yet close cut-points would roughly be the same for an expert, yet would be maximally different in this paper. Hence, I would like authors to clarify how much they think this distance is natural, and how much is it a convenient way to get theoretical results?

* Maybe some intuition about the theorems would be good. For example, from the text alone, it appears that theorems 4.1. and 5.1. are exactly the same (at least I could not detect any textual/formal difference). Similarly, it is not clear at all whether Theorem 6.1. is in general useful, as the optimal stump making only one cut may actually have a very bad accuracy (especially at startup)?

* From a non-expert view, it is very difficult to know whether the obtained theoretical results are easy derivation from previous results in the same vein, or are genuinely difficult to obtain. In connection with the above remark, it is also quite difficult to see how strongly connected those theoretical results are to the actual practical implementation/methodology, and helps in predicting the behaviours of those? Maybe a bit more intuition would be needed to fully appreciate this.

**Summary Of The Paper:**

The paper describes a mean to lower the high sensibility of decision trees using randomization of algorithm construction. Some theorems are given and experiments are done showing that in general, trees obtained by the randomization process are less sensitive.

**Summary Of The Review:**

The paper requires one to know quite well the particular approaches used in the paper (derived by Yoshida and colleagues, for the most part of it), that seems to be well-known in combinatorial optimisation, maybe a bit less in machine learning. As I had to make an emergency review, I could just skim over those and my familiarity with them is too poor to make a strong judgment about the used techniques.

All in all, the approach looks interesting, but it would require a bit of clarifications for non-expert readers to better follow the paper. In its current form, intuition is sometimes hard to grasp, but the methodology is described with sufficient details to be reproducible. There are some aspects in the experiments that I could not interpret given my understanding, however.

---

> ### Author Response · Authors · 2022-11-08
> **Reply to the review (1/2)**
>
> We thank the reviewer for providing helpful comments.
>
> > If I understand Algorithm 3 well, each potential cut or optimal split of the tree is now a probabilistic decision (the cut is made or not). I honestly do not understand how this result in more stability of the tree? In particular, one would expect randomized cuts to put more diversity in built trees? What is the general value of the probability of accepting a cut (it is hard to grasp intuitively at first sight without being familiar with the approach). For example, in Figure 3.b, how does randomization help to have more systematic obtention of the original tree in the second line?
>
> A deterministic algorithm tends to have a high sensitivity because its decisions are discrete, and hence a small perturbation to the input data may cause a significant change in the output. By contrast, the distribution of the output of a randomized algorithms can continuously change as the input slightly change, and hence is more stable.
>
> Below, we explain the intuition using a simple example. Suppose that we have two candidate rules $\omega_1, \omega_2$ with their optimal scores $\mathrm{opt}\_{\omega_1, B}(\mathcal{L}) = 90$, $\mathrm{opt}\_{\omega_2, B}(\mathcal{L}) = 89$. The greedy algorithm selects $\omega_1$ with the largest score. Now, suppose the scores have changed after removing a subset $\mathcal{S} \subset \mathcal{L}$ as $\mathrm{opt}\_{\omega_2, B}(\mathcal{L} \setminus \mathcal{S}) = 85$, $\mathrm{opt}\_{\omega_2, B}(\mathcal{L} \setminus \mathcal{S}) = 86$. In this case, the greedy algorithm selects $\omega_2$ with the largest score. Hence, there is zero probabily of having the same trees for $\mathcal{L}$ and $\mathcal{L} \setminus \mathcal{S}$. This is not the case for the proposed algorithm. For both $\mathcal{L}$ and $\mathcal{L} \setminus \mathcal{S}$, the optimal scores of $\omega_1$ and $\omega_2$ are very close, inducing their selection probabilties $\mathrm{Pr}(\text{select } \omega_1) \approx \mathrm{Pr}(\text{select } \omega_2) \approx 0.5$. That is, the output distributions of the proposed algorithm for $\mathcal{L}$ and $\mathcal{L} \setminus \mathcal{S}$ are close.
>
> To produce Figure 3.b, we need to guarantee that the output trees for $\mathcal{L}$ and $\mathcal{L} \setminus \mathcal{S}$ are close (with high probability over the randomness in the algorithm). For this purpose, we used the algorithm given in Section 5 with a low expected deterministic average sensitivity. In the example above, if we train trees independently for $\mathcal{L}$ and $\mathcal{L} \setminus \mathcal{S}$, the resulting trees will be different; the chosen rules to be the same with probability approximately 0.5 because $\mathrm{Pr}(\text{select } \omega_1 \text{ in } \mathcal{L} \land \text{select } \omega_1 \text{ in } \mathcal{L} \setminus \mathcal{S}) \approx 0.25$ and same for $\omega_2$. However, if we use the same random bits $\pi$ for both $\mathcal{L}$ and $\mathcal{L} \setminus \mathcal{S}$, we have a larger probability of having the same trees; Lemma C.1 confirms that  $\mathrm{Pr}(\text{select same } \omega \text{ in } \mathcal{L} \text{ and } \mathcal{L} \setminus \mathcal{S})$ can be close to one if the scores of $\omega_1$ and $\omega_2$ are close both in $\mathcal{L}$ and $\mathcal{L} \setminus \mathcal{S}$.
>
>
> > It would be good to have some idea of the variance values over some of the graphs. For example, in Figure 2.c, one would expect the high variations to be due to variance and not general trends, especially as those looks very small and edgy?
>
> Following the suggestion, we added figures with errors in Appendix D.1 in the revised version.
>
> > In figure 4, it seems that the evolution of all the curves in terms of training accuracy is the same for all the settings (for each curve, the sequence of ordinate values are exactly the same. Given the fact that this are randomized splits over different data sets, how is this possible?
>
> The training accuracy in the vertical axis is the accuracy of the tree trained using the whole sampled training set without data removal. Because this accuracy is independent from the number of removed data points, the plots share exactly the same accuracy.

---

> ### Author Response · Authors · 2022-11-08
> **Reply to the review (2/2)**
>
> > I would disagree to some extent that a distance based on the maximal common sub-tree is a natural distance between learned decision trees. In particular, it seems the distance will consider that a split made in the same variable but on a possibly slightly different value will lead to different trees, which is highly questionable interpretability-wise, as two trees with all the same variables but different yet close cut-points would roughly be the same for an expert, yet would be maximally different in this paper. Hence, I would like authors to clarify how much they think this distance is natural, and how much is it a convenient way to get theoretical results?
>
> The reason that we adopt the current one is that it is a pessimistic distance measure, and hence the obtained theoretical bound can also be used to bound the one that incorporates similarity of cut-points. We have added empirical results for a relaxed distance that ignores cut-points in Appendix D.1 in the revised version. We can confirm that it has a negligible impact.
>
> > Maybe some intuition about the theorems would be good. For example, from the text alone, it appears that theorems 4.1. and 5.1. are exactly the same (at least I could not detect any textual/formal difference). Similarly, it is not clear at all whether Theorem 6.1. is in general useful, as the optimal stump making only one cut may actually have a very bad accuracy (especially at startup)?
>
> The difference between Theorems 4.1 and 5.1 is that the former bounds the average sensitivity whereas the latter bounds the expected deterministic average sensitivity, and bounding the latter is more useful in practice because we obtain similar decision trees using the same random seed. We agree that the accuracy guarantee of Theorem 6.1 is not strong, but it is somehow the best we can hope for because existing polynomial-time decision tree learning algorithms have similar accuracy guarantees.
>
> > From a non-expert view, it is very difficult to know whether the obtained theoretical results are easy derivation from previous results in the same vein, or are genuinely difficult to obtain. In connection with the above remark, it is also quite difficult to see how strongly connected those theoretical results are to the actual practical implementation/methodology, and helps in predicting the behaviours of those? Maybe a bit more intuition would be needed to fully appreciate this.
>
> The idea of using the exponential mechanism to stabilize algorithms is not very new (for example, it appeared in the differential privacy literature), but using it in a recursive fashion to obtain sensitivity bounds for the tree distance is novel. Also bounding expected deterministic sensitivity required some trick such as rejection sampling (Lines 4 to 7 in Algorithm 4).

---

### Official Review · Reviewer_s724 · 2022-10-31

**Confidence:** 3
**Correctness:** 4
**Technical Novelty And Significance:** 3
**Empirical Novelty And Significance:** 3
**Recommendation:** 6

**Clarity, Quality, Novelty And Reproducibility:**

Clarity. Most of the parts of the paper are clearly written. A weak point is a missing intuitive explanation of the proposed method (Algorithm 3). The algorithm differs from the standard greedy method by introducing randomness in the selection of the decision rule, which results in increasing the stability (i.e. decreasing AS). This is quite counter-intuitive on its own. The implementation of the randomized strategy, which was at least for me important to understand how the stability is introduced, is deferred to the appendix. I believe more space in the paper should be dedicated to explaining how stability can emerge from the randomized learning strategy.

Typos:
- It is not mentioned that Theorem 4.1. applies to Algorithm 3.
- pp 5: "eseentially"
- pp 5: Typos in the definition of the set $\Omega$.

Novelty. To my knowledge, the application of the AS as a measure of the stability of the decision tree learning algorithm is novel.

Quality. The paper seems technically sound. Statements are mathematically precise and supported by proofs.

Reproducibility. The authors provide code that allows to reproduce the results.

**Strength And Weaknesses:**

Strengths:

The notion of stability of the learning algorithm is both practically relevant and theoretically interesting. There are not many existing papers on the topic. This paper provides a way how to study the issue of stability of decision trees formally.

The authors prove approximation guarantees for the proposed algorithm and the algorithm's ability to decrease the AS.

The method is relatively simple modification of the standard greedy algorithm.

Weaknesses:

Although AS is exactly defined, it is not clear how relevant this measure is in practical applications. E.g. it would be more appealing to know a probability with which the learned tree is exactly the same after perturbing the data, or a probability that AS will below a specified threshold. On the other hand currently there are not much other alternatives.

The proven bounds show that AS of the proposed algorithm decreases with increasing number of examples. However, the bounds do not show why the proposed method should outperform the standard greedy algorithm. On the other hand, experimental results suggest that on small data the proposed method is indeed beneficial if compared to the greedy one.

**Summary Of The Paper:**

[Varma et al 2021] recently proposed the notion of average sensitivity to measure the stability of solutions produced by graph algorithms. The authors of the paper under review propose to use the AS as a measure of the stability of algorithms learning decision trees w.r.t. random permutations of the training examples. The authors further propose an algorithm whose average sensitivity can be decreased at the cost of decreasing the prediction accuracy. The method is empirically evaluated and compared against standard greedy algorithm.

**Summary Of The Review:**

The paper introduces the notion of average sensitivity as a measure of the stability of decision tree algorithms and proposes a simple randomized strategy to reduce it. Experiments show benefits of the proposed randomized method over standard greedy algorithm on small sample sizes. The paper would benefit from adding an intuitive explanation of the key idea, i.e. increasing stability of the learning algorithm via introducing randomness.

---

> ### Author Response · Authors · 2022-11-07
> **Clarification on the comments**
>
> We thank the reviewer for helpful comments. We are now preparing the replies to the comments.
> For us to prepare appropriate replies, we would like to ask you for the clarification of a couple of your comments.
>
> > The proven bounds show that AS decreases with increasing number of examples. Therefore, the proposed algorithm is likely to be beneficial over a standard greed algorithm only when the number of examples is small. Unfortunately, the authors compared the methods only on small data (<=1000 examples).
>
> As pointed out, the bound becomes small for large n, i.e., for large datasets. That is, the proposed algorithm is advantageous for large datasets. Perhaps, you may intended to say "the proposed algorithm is likely to be beneficial over a standard greed algorithm only when the number of examples is **large**"?
>
> >  it is unclear whether AS has a clear interpretation that would be relevant for practical applications and whether the proposed algorithm is beneficial for other than small datasets.
>
> The same question goes here. Perhaps, you may intended to say "whether the proposed algorithm is beneficial for other than **large** datasets"?

---

> > ### Comment · Reviewer_s724 · 2022-11-07
> > **Reply to the authors' question on the clarification of my comment.**
> >
> > I may be wrong, however it seems to me that for large n the stability of the standard greedy algorithm will be ok (removing examples will have only a small effect on the results), and hence there will be actually no need to use the proposed randomized algorithm. Hence I was wondering whether the proposed method is beneficial over the greedy algorithm when n is larger. Or, whether it is useful only for small datasets when stability is an issue. Does it make sense?

---

> > > ### Author Response · Authors · 2022-11-08
> > > **Thank you very much for you clarification.**
> > >
> > > Thank you very much for you clarification.
> > > Please find our reply to the review in a separate post.

---

> ### Author Response · Authors · 2022-11-08
> **Reply to the review**
>
> We thank the reviewer for helpful comments. We fixed the typos in the updated manuscript. We provide detailed answers to other comments below.
>
> > Although it is exactly defined, it is not clear whether AS has a clear interpretation that would be relevant for some practical application of decision trees. If one would know that algorithm X has the average sensitivity Y, how can it be used?
>
> Because we can compare algorithms using AS, we can select the one with a smaller AS (if its accuracy is reasonably good). Without AS, we cannot argue which one is superior to which.
>
>
> > it seems to me that for large n the stability of the standard greedy algorithm will be ok (removing examples will have only a small effect on the results), and hence there will be actually no need to use the proposed randomized algorithm. Hence I was wondering whether the proposed method is beneficial over the greedy algorithm when n is larger. Or, whether it is useful only for small datasets when stability is an issue.
>
> We agree that for the large-data regime, the greedy algorithm can be sufficiently stable. We therefore think the proposed algorithm is especially useful in the small-data regime. Our empirical results confirm that our algorithms work well in practice for such small data.  Considering the fact that many data used in industries, such as those obtained from medical diagnosis and physical/chemical experiments, are small, we believe that providing stable algorithms is very important for practitioners working on small data.
>
> > A weak point is a missing intuitive explanation of the proposed algorithm (Algo 3). The algorithm differs from the standard greedy method by introducing randomness in the selection of the decision rule, which results in increasing the robustness (decreasing AC). This is quite counterintuitive (randomness increases stability) on its own and deserves an explanation
>
> A deterministic algorithm tends to have a high sensitivity because its decisions are discrete, and hence a small perturbation to the input data may cause a significant change in the output. By contrast, the distribution of the output of a randomized algorithms can continuously change as the input slightly change.
>
> Below, we explain the intuition using a simple example. Suppose that we have two candidate rules $\omega_1, \omega_2$ with their optimal scores $\mathrm{opt}\_{\omega_1, B}(\mathcal{L}) = 90$, $\mathrm{opt}\_{\omega_2, B}(\mathcal{L}) = 89$. The greedy algorithm selects $\omega_1$ with the largest score. Now, suppose the scores have changed after removing a subset $\mathcal{S} \subset \mathcal{L}$ as $\mathrm{opt}\_{\omega_2, B}(\mathcal{L} \setminus \mathcal{S}) = 85$, $\mathrm{opt}\_{\omega_2, B}(\mathcal{L} \setminus \mathcal{S}) = 86$. Then, the greedy algorithm selects $\omega_2$ because it has now the largest score. Hence, there is zero probability of having the same trees for $\mathcal{L}$ and $\mathcal{L} \setminus \mathcal{S}$. This is not the case for the proposed (randomized) algorithm. For both $\mathcal{L}$ and $\mathcal{L} \setminus \mathcal{S}$, the optimal scores of $\omega_1$ and $\omega_2$ are very close, inducing their selection probabilities $\mathrm{Pr}(\text{select } \omega_1) \approx \mathrm{Pr}(\text{select } \omega_2) \approx 0.5$. That is, the output distributions of the proposed algorithm for $\mathcal{L}$ and $\mathcal{L} \setminus \mathcal{S}$ are close.
>
>
> > some issues are not discussed, e.g., how to setup the "epsilon" parameter in practice.
>
> There is no optimal $\epsilon$ in general because the user should tune it considering the trade-off between average sensitivity and accuracy according to his/her own demands (similar to differential privacy where the privacy level is determined by the user or by some external requirements).

---

### Decision · Program_Chairs · 2023-01-20

**Decision:**

Accept: poster

**Justification For Why Not Higher Score:**

The paper presents an interesting result, but its scope is rather limited and interesting for researchers working on specific models and methods.

**Justification For Why Not Lower Score:**

This paper can motivate interesting research around interpretable models such as decision trees and rule-based systems. It also nicely links theoretical computer science with machine learning. During the AC-reviewer meeting we decided to move forward with this submissions as the weaknesses of the paper can be easily fixed, but the paper can be influential for researchers working on this type of models and methods.

**Metareview: Summary, Strengths And Weaknesses:**

The paper gives an interesting insight into learning algorithms that provide more "stable" decision trees. The introduced algorithm in a smart way picks a cut that should be more robust to small perturbations in training data than the typical greedy one. Interestingly, this algorithm uses randomized cuts, but the way they are obtained ensures that in expectation the resulting trees are more stable. The authors present the results in a rigorous way by formally proving their main results. Additional experimental studies illustrate the main theoretical findings.

The reviewers and AC are leaning towards accepting the paper. Nevertheless, we ask the authors to improve the paper in the following points:
- Discussion on guarantees for the greedy method and the small sample regime: The theoretical gap between the greedy method and the introduced one should be discussed in the main text,
- Moving the description/algorithm of the StableDR method to the main paper (at least some headlines should be given in the main text): The method by itself nicely explains why a randomized cut can improve stability; any additional intuitive examples including those given in the rebuttal shall also be considered for adding to the main text,
- Guidelines for practitioners on how to trade-off both accuracy and stability: a short discussion on the intuition how to find the right trade-off between accuracy and stability in a given application should also be considered for adding to the main text.

**Note From Pc:**

if the above contains the word "oral" or "spotlight" please see: "oral" presentation means -> notable-top-5% and "spotlight" means -> notable-top-25%. As stated in our emails, we are disassociating presentation type from AC recommendations

**Summary Of Ac-Reviewer Meeting:**

We met with two out of 4 reviewers. The main problem were different time zones. The reviewers that could not attend the meeting shared their opinions via email and/or OpenReview.

We mainly discussed the three points mentioned in the meta-review:
- Guarantees for the greedy method and the small sample regime: This part should be better presented in the main text to show why the greedy algorithm fails in the small sample regime.
- Moving the description of the StableDR method to the main paper: The algorithm by itself gives an intuition why the theoretical claims are correct.
- Guidelines for practitioners on how to trade-off both accuracy and stability: This is a common problem when using multiple criteria for evaluation. Therefore the authors should try to give some hints how to properly build an intuition to find the right trade-off in a given application.